# NEUMATC: A GENERAL NEURAL FRAMEWORK FOR FAST PARAMETRIC MATRIX OPERATION

## ABSTRACT

Matrix operations (e.g., inversion and singular value decomposition (SVD)) are fundamental in science and engineering. In many emerging real-world applications (such as wireless communication and signal processing), these operations must be performed repeatedly over matrices with parameters varying continuously. However, conventional methods tackle each matrix operation independently, underexploring the inherent low-rankness and continuity along the parameter dimension, resulting in significantly redundant computation. To address this challenge, we propose *Neural Matrix Computation Framework* (**NeuMatC**), which elegantly tackles general parametric matrix operation tasks by leveraging the underlying low-rankness and continuity along the parameter dimension. Specifically, NeuMatC unsupervisedly learns a low-rank and continuous mapping from parameters to their corresponding matrix operation results. Once trained, NeuMatC enables efficient computations at arbitrary parameters using only a few basic operations (e.g., matrix multiplications and nonlinear activations), significantly reducing redundant computations. Experimental results on both synthetic and real-world datasets demonstrate the promising performance of NeuMatC, exemplified by over $3\times$ speedup in parametric inversion and $10\times$ speedup in parametric SVD compared to the widely used NumPy baseline in wireless communication, while maintaining acceptable accuracy.

## 1 INTRODUCTION

Neural networks have emerged as a promising tool in scientific computing, enabling data-driven solutions to classical numerical problems (Kopaničáková & Karniadakis, 2025; Karniadakis et al., 2021; Bhattacharya et al., 2021; Indyk et al., 2019). For example, recent works have demonstrated its effectiveness in solving partial differential equations by physics-informed neural networks (Cai et al., 2021; Kashinath et al., 2021; Gao et al., 2023), as well as in spectral problems such as eigenfunction approximation and high-dimensional eigenvalue computation (Han et al., 2020; Zhang et al., 2021). Additionally, reinforcement learning has been explored for discovering efficient matrix product schemes (Fawzi et al., 2022). These advances highlight the capability of neural networks in tackling core numerical tasks traditionally dominated by analytical and numerical approaches.

A fundamental component in scientific computing is matrix operations, including inversion, eigenvalue decomposition, and singular value decomposition (SVD) (Ji et al., 2025; Park & Nakatsukasa, 2025). Matrix operations have long attracted significant research interest, leading to the development of a variety of methods. Classical methods for matrix operations include direct solvers (e.g., Gaussian elimination with LU for matrix inversion and QR-based algorithms for SVD) and iterative methods including the conjugate gradient method and inverse approximations based on the Neumann series (Zhu et al., 2015; Hestenes et al., 1952). These methods are widely valued for their precision, stability, and strong theoretical guarantees (Golub & Loan, 1996). More recently, randomized methods have received increasing attention for accelerating matrix operations (Halko et al., 2011; Kaloorazi & Chen, 2020; Kaloorazi & de Lamare, 2018). By introducing random sampling to extract informative matrix components, these methods reduce the computational cost of operations while maintaining acceptable accuracy. However, despite their precision and well-established theoretical guarantees, classical numerical and randomized methods rely on step-by-step factorization or iterative refinement, which are inherently sequential and thus difficult to fully parallelize on modern platforms (Dongarra et al., 1991). In contrast, deep learning offers a promising alternative, leveraging the strengths of modern hardware to develop efficient methods for matrix operations.

Figure 1: Comparison between conventional numerical methods and the proposed NEUMATC. Conventional methods tackle each matrix operation independently, leading to redundant computation. In contrast, NEUMATC learns a low-rank and continuous mapping from parameters to the corresponding operation results using only a few basic operations (i.e., matrix multiplications and nonlinear activations). *Validated on real wireless communication scenarios,* NEUMATC *achieves over* $3\times$ *speedup in parametric inversion and* $10\times$ *speedup in parametric SVD compared to the widely used NumPy baseline, while maintaining acceptable accuracy (see Section 3).*

While matrix operations play a foundational role in scientific computing and engineering applications, the potential of neural networks for these tasks remains largely underexplored. Among existing works, one research direction integrates neural networks into classical iterative solvers to improve numerical performance(Almasadeh et al., 2022; Feng et al., 2001; Li & Hu, 2022). For example, Li and Hu proposed a second-order neural network framework that incorporates Newton-type updates (Li & Hu, 2022). Feng *et al.* proposed a recurrent cross-associative neural network for computing the matrix SVD (Feng et al., 2001). However, these methods provide limited speedups compared to highly optimized numerical libraries such as LAPACK (Anderson et al., 1999). Another line of research focuses on end-to-end models that directly regress the result of matrix operations from input data. For example, Ji *et al.* proposed a fully connected network that directly maps flattened matrices to their inverses (Ji et al., 2025). However, these methods suffer from scalability issues: as the input matrix grows in size, the corresponding network becomes increasingly large and leads to slower inference speed.

In emerging real-world applications (e.g., wireless communication, signal processing, and control systems), matrix operations are not applied to a single matrix, but on a family of matrices parameterized by $p$, where $p$ (e.g., frequency, location, or time) varies continuously (Wu et al., 2025; Brandt & Bengtsson, 2011; Wilber et al., 2022; Park & Nakatsukasa, 2025; Zahm & Nouy, 2016). These parametric computation scenarios further expose a key shortcoming of conventional methods for matrix operations: existing methods *typically tackle each matrix independently, ignoring the continuity and the low-rank structure along the parameter dimension, which leads to significant computational redundancy.* This naturally raises an important question:

*How to efficiently perform matrix operations over parametric matrices by leveraging the inherent continuity and low-rankness?*

To address this challenge, we propose ***Neural Matrix Computation Framework (NeuMatC)***, which tackles general parametric matrix operations by leveraging the underlying low-rankness and continuity along the parameter dimension. Specifically, NeuMatC unsupervisedly learns a low-rank and continuous mapping from parameters to their corresponding operation results. Once trained, NeuMatC enables efficient computations at arbitrary parameters using only a few basic operations (e.g., matrix multiplications and nonlinear activations), significantly reducing redundant computations required by conventional methods. As illustrated in Figure 1, rather than computing matrices independently at each parameter, NeuMatC learns a low-rank and continuous mapping that directly maps parameters to the corresponding matrix operation results. Experimental results on both synthetic and real-world datasets demonstrate the promising performance of NeuMatC, exemplified by over $3\times$ speedup in parametric inversion and $10\times$ speedup in parametric SVD compared to the widely used NumPy baseline in wireless communication, while maintaining acceptable accuracy.

The remainder of this paper is organized as follows: Section 2 details the NeuMatC framework. Section 3 presents experimental results. In Section 4, we discuss key aspects of the proposed NeuMatC. Finally, Section 5 concludes the paper.

## 1.1 RELATED WORK

Despite the importance of parametric matrix computation in emerging applications, this area has received relatively limited attention. A representative class is based on ***Zhang Neural Networks (ZNN)*** (Chen & Zhang, 2020; Zhang & Yi, 2011; Guo et al., 2017; Zhang et al., 2008; Jin et al., 2022), which formulate matrix operations as differential equations whose equilibria correspond to the operation results, and solve them sequentially along the parameter dimension. However, ZNN processes each parameter point sequentially, which is inefficient for many emerging real-world applications (e.g., channel precoding and detection) that require fast computation across a large number of sampled parameter values. Moreover, each step of ZNN entails a relatively high computational cost due to integration. Several methods aim to ***accelerate the solution of parametric linear systems*** (Correnty et al., 2024; Kopaničáková & Karniadakis, 2025; Jarlebring & Correnty, 2022). Traditional solvers such as inexact infinite GMRES (Correnty et al., 2024) offer efficient Krylov-based strategies but require iterative processing at each parameter point. Recent hybrid methods, such as DeepONet preconditioners (Kopaničáková & Karniadakis, 2025), combine operator learning with Krylov solvers. However, these methods are operation-specific and rely on solver-in-the-loop inference. In contrast, NeuMatC is a general neural framework for fast parametric matrix computation that can handle a broad class of matrix operations (including inversion, eigenvalue decomposition, and SVD). By learning a low-rank and continuous mapping, NeuMatC can efficiently compute the operation results at arbitrary parameters using only a few basic operations (e.g., matrix multiplications and nonlinear activations).

## 2 NEURAL MATRIX COMPUTATION FRAMEWORK (NEUMATC)

In this section, we present the NeuMatC framework for fast parametric matrix operations. At the core of NeuMatC is ***the efficient mapping*** from parameters to matrix operation results, which fully leverages the inherent low-rankness and continuity along the parameter dimension. To ***effectively learn this mapping***, we design an algebraic structure-aware loss function, along with a failure-informed adaptive sampling strategy.

**Notations.** Throughout this work, we adopt standard notation: lowercase letters $a \in \mathbb{R}$ denote scalars, bold lowercase letters $\mathbf{a} \in \mathbb{R}^{n_1}$ denote vectors, bold uppercase letters $\mathbf{A} \in \mathbb{R}^{n_1 \times n_2}$ denote matrices, and calligraphic letters $\mathcal{A} \in \mathbb{R}^{n_1 \times n_2 \times n_3}$ denote third-order tensors. We consider a continuous parametric matrix $\mathbf{A}(p) \in \mathbb{R}^{n_1 \times n_2}$, where $p \in \mathbb{P}$ and $\mathbb{P} \subseteq \mathbb{R}$ is the parameter domain. Applying a matrix operation to $\mathbf{A}(p)$ gives an output of the form $\mathbf{G}(p) = \{\mathbf{G}_1(p), \cdots, \mathbf{G}_m(p)\}$, where each $\mathbf{G}_i(p) \in \mathbb{R}^{n_{1i} \times n_{2i}}$ denotes a component of the result. For example, matrix inversion corresponds to $\mathbf{G}(p) = \mathbf{A}(p)^{-1}$ and SVD corresponds to $\mathbf{G}(p) = \{\mathbf{U}(p), \mathbf{S}(p), \mathbf{V}(p)\}$ such that $\mathbf{A}(p) = \mathbf{U}(p)\mathbf{S}(p)\mathbf{V}(p)^\top$. More definitions are provided in Appendix A.

### 2.1 LOW-RANK AND CONTINUOUS MAPPING

In NeuMatC, we aim to learn an efficient mapping from parameters to the corresponding matrix operation results. However, directly learning such a mapping can be computationally inefficient and generalize poorly. To overcome this limitation, NeuMatC leverages two general properties commonly observed in real-world parametric matrix operation problems: ***low-rankness*** and ***continuity*** of the operation results along the parameter dimension. Empirical evidence indicates that the results of many matrix operations in real-world scenarios often lie in low-dimensional subspaces along the parameter dimension, as demonstrated by the rapid decay of singular values (see Fig. 2). Moreover, for a continuous parametric matrix $\mathbf{A}(p)$, many standard matrix operations (e.g., inversion and SVD) preserve continuity, i.e., the corresponding operation results also vary continuously with

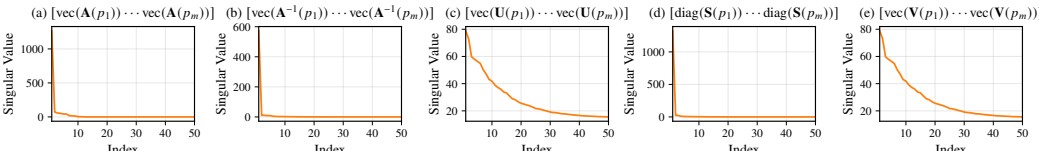

Figure 2: Singular value decay of matrices constructed by stacking vectorized forms of $\mathbf{A}(p)$, $\mathbf{A}^{-1}(p)$, and their SVD components $\mathbf{U}(p)$, $\mathbf{S}(p)$, and $\mathbf{V}(p)$ across $m$ parameter instances. The rapid decay of singular values indicates low-rank structure along the parameter dimension.

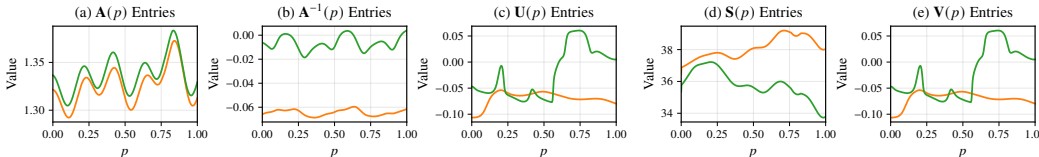

Figure 3: Representative sampled entries of a real-world channel matrix $\mathbf{A}(p)$, its inverse $\mathbf{A}(p)^{-1}$, and its SVD outputs (singular values $\mathbf{S}(p)$ and left and right singular vectors $\mathbf{U}(p)$ and $\mathbf{V}(p)$) along the parameter dimension $p$, demonstrating the continuity of these matrices. The green and orange curves correspond to two different sampled entries

the parameter $p$ (see Fig. 3). This continuity property can be theoretically established under mild conditions (see Appendix B).

The low-rank structure along the parameter dimension observed in parametric matrix operation results $\mathbf{G}(p) = \{\mathbf{G}_1(p), \cdots, \mathbf{G}_m(p)\}$ motivates the following compact mapping for each component. Specifically, we define $\widetilde{\mathbf{G}}_i : \mathbb{R} \to \mathbb{R}^{n_{1i} \times n_{2i}}$, which maps a parameter value $p$ to the corresponding operation result component $\mathbf{G}_i(p)$:

$$\widetilde{\mathbf{G}}_i(\cdot) := \mathcal{C}_i \times_3 \Phi_i(\cdot), \tag{1}$$

where $\mathcal{C}_i \in \mathbb{R}^{n_{1i} \times n_{2i} \times d_i}$ is a latent tensor and $\Phi : \mathbb{R} \to \mathbb{R}^{d_i}$ is a vector-valued function. $d_i$ denotes the latent dimension and $\times_3$ denotes the mode-3 matrix–tensor product (see Appendix A for details). A detailed discussion of the low-rank structure particularly for the case of matrix inversion is provided in Appendix E.

The following theorem provides a theoretical justification for this compact mapping design, showing that under mild conditions, the operation result component $\mathbf{G}_i(p)$ at arbitrary parameter values $p$ can be obtained by the mapping $\widetilde{\mathbf{G}}_i(\cdot) = \mathcal{C}_i \times_3 \Phi_i(\cdot)$; the proof is provided in Appendix C.

**Theorem 1** (Existence of the Compact Mapping). *Let* $\mathbf{G}(p) = \{\mathbf{G}_1(p), \cdots, \mathbf{G}_m(p)\}$ *be a parametric matrix operation result, where each* $\mathbf{G}_i(p) \in \mathbb{R}^{n_{1i} \times n_{2i}}$ *for* $i = 1, \cdots, m$. *Assume that for each* $i$, *the set* $\{[\mathbf{G}_i(p)]_{jk} \mid j = 1, \cdots, n_{1i}, \ k = 1, \cdots, n_{2i}\}$ *spans a* $d_i$-*dimensional linear subspace, where* $[\mathbf{G}_i(p)]_{jk}$ *denotes the* $(j, k)$-*th entry of* $\mathbf{G}_i(p)$. *Then, there exists a latent tensor* $\mathcal{C}_i \in \mathbb{R}^{n_{1i} \times n_{2i} \times d_i}$ *and a vector-valued function* $\Phi_i : \mathbb{R} \to \mathbb{R}^{d_i}$ *such that the following mapping*

$$\widetilde{\mathbf{G}}_i(\cdot) := \mathcal{C}_i \times_3 \Phi_i(\cdot)$$

*satisfies* $\widetilde{\mathbf{G}}_i(p) = \mathbf{G}_i(p)$ *for all* $p \in \mathbb{P}$.

Selecting an appropriate $\Phi_i(\cdot)$ is crucial to the effectiveness of the proposed compact mapping. In NeuMatC, we construct the function $\Phi_i(\cdot)$ using a multilayer perceptron (MLP) $\Phi_{\theta_i}(\cdot)$, considering its strong effectiveness capability. The mapping of each output component is then given by $\widetilde{\mathbf{G}}_i(\cdot) = \mathcal{C}_i \times_3 \Phi_{\theta_i}(\cdot)$, where $\mathcal{C}_i$ is a learnable latent tensor and $\theta_i$ denotes the MLP parameters.

*By exploiting the low-rankness along the parameter dimension, the suggested compact mapping enables efficient computation for arbitrary parameters, requiring only a few basic operations (i.e., matrix multiplications and nonlinear activations).*

**Remark 1.** *Traditional low-rank methods such as randomized SVD typically rely on a strong assumption that each individual matrix is low-rank. In contrast, NeuMatC does not assume low-rankness of the individual matrices. We adopt a* milder *assumption commonly observed in real-world applications, i.e., low-rankness along the* parameter *dimension. That is the collection* $\{\mathbf{G}(p) : p \in \mathcal{P}\}$ *approximately lies in a low-dimensional subspace of* $\mathbb{R}^{n \times n}$.

**Continuity in NeuMatC.** Another important property commonly observed in parametric matrix operation results is *continuity*. In this part, we demonstrate that, in addition to capturing the low-rank structure, the proposed NeuMatC framework also captures this continuity. Theoretical justification is provided in the following theorem.

**Theorem 2** (Lipschitz Continuity in NeuMatC). *Given the mapping* $\widetilde{\mathbf{G}}_i(\cdot) = \mathcal{C}_i \times_3 \Phi_{\theta_i}(\cdot)$, *where* $\mathcal{C}_i \in \mathbb{R}^{n_{1i} \times n_{2i} \times d_i}$ *is a latent tensor and* $\Phi_{\theta_i}(\cdot) : \mathbb{R} \to \mathbb{R}^{d_i}$ *is an* $L$-*layer MLP with Lipschitz-continuous activation function* $\sigma(\cdot)$ *with Lipschitz constant* $L_\sigma$. *Assume that* $\|\mathcal{C}_i\|_1 \leq \kappa$ *and each weight matrix* $\mathbf{W}_\ell$ *in the MLP satisfies* $\|\mathbf{W}_\ell\|_1 \leq \eta$. *Then for any* $p_1, p_2 \in \mathbb{R}$, *the output satisfies the bound*

$$\|\widetilde{\mathbf{G}}_i(p_1) - \widetilde{\mathbf{G}}_i(p_2)\|_F \leq \kappa (L_\sigma \eta)^L \cdot |p_1 - p_2|. \tag{2}$$

The proof of Theorem 2 is provided in Appendix D.

**Remark 2.** *Theorem 2 reveals that the continuity of the proposed mapping depends on the Lipschitz constant of the activation function. In practice, we use sine activations $\sigma(\cdot) = \sin(\omega\cdot)$, which are Lipschitz continuous. The frequency parameter $\omega$ offers a direct control over output continuity. Smaller $\omega$ leads to smoother variation, while larger $\omega$ allows the model to capture rapid changes.*

*In summary, the proposed mapping harmoniously exploits **both the low-rankness and the continuity** of matrix operation results along the parameter dimension, enabling **both efficient and effective computation** over the parameter domain.*

## 2.2 TRAINING OF NEUMATC

In this section, we present how NeuMatC is trained to effectively learn the low-rank and continuous mapping. Specifically, NeuMatC is trained in an unsupervised manner using only a few samples to learn the mapping. Given limited samples, we design ***an algebraic structure-aware loss function***, which encourages algebraic structure consistency at a set of unsupervised collocation points. Correspondingly, we further introduce ***failure-informed adaptive sampling*** to select informative collocation points to improve generalization.

**Algebraic Structure-Aware Loss Function.** In this part, we design the loss function of NeuMatC, which enables learning from limited samples by enforcing algebraic structure consistency. Suppose we are given a parametric matrix $\mathbf{A}(p) \in \mathbb{R}^{n_1 \times n_2}$ that varies continuously with parameter $p$. The set of supervised sampling points is denoted by $\mathcal{D}_{\text{data}} = \{(p_j, \mathbf{G}(p_j))\}_{j=1}^{N_s}$, where each $\mathbf{G}(p_j)$ denotes a ground-truth output computed by conventional numerical solvers. To further alleviate reliance on a high sampling rate of supervised sampling points, we introduce a separate set of unsupervised collocation points $\mathcal{D}_{\text{col}} = \{q_\ell\}_{\ell=1}^{N_c}$, at which we enforce consistency with the algebraic structure of the matrix operation without access to ground-truth outputs. Using the MLP-parameterized representation, we formulate the following loss function:

$$\min_{\{\mathcal{C}_i, \theta_i\}_{i=1}^m} \underbrace{\sum_{i=1}^m \sum_{j=1}^{N_s} \|\mathcal{C}_i \times_3 \Phi_{\theta_i}(p_j) - \mathbf{G}_i(p_j)\|_F^2}_{\text{Data Fidelity Loss}} + \lambda \underbrace{\sum_{\ell=1}^{N_c} \|\mathscr{R}(q_\ell)\|_F^2}_{\text{Structure Consistency Loss}}, \tag{3}$$

where $\lambda$ is a trade-off parameter. The residual $\mathscr{R}(\cdot)$ measures the violation of the underlying algebraic structure and depends on the specific matrix operation. For example, in the case of matrix inversion where the target satisfies $\mathbf{A}(p)\mathbf{G}(p) = \mathbf{I}$, the residual is $\mathscr{R}(p) = \mathbf{A}(p) \cdot (\mathcal{C} \times_3 \Phi_\theta(p)) - \mathbf{I}$. For SVD, the residual $\mathscr{R}(p)$ includes both the reconstruction error $\mathbf{A}(p) - \widetilde{\mathbf{U}}(p)\widetilde{\mathbf{S}}(p)\widetilde{\mathbf{V}}(p)^\top$ and the orthogonality violations $\widetilde{\mathbf{U}}(p)^\top \widetilde{\mathbf{U}}(p) - \mathbf{I}$ and $\widetilde{\mathbf{V}}(p)^\top \widetilde{\mathbf{V}}(p) - \mathbf{I}$, where $\widetilde{\mathbf{G}}(p) = \{\widetilde{\mathbf{U}}(p), \widetilde{\mathbf{S}}(p), \widetilde{\mathbf{V}}(p)\}$ is the SVD output of NeuMatC. *The structure consistency loss reduces reliance on a high sampling rate of supervised samples across the parameter domain.*

**Remark 3** (Connection to Physics-Informed Neural Networks (PINNs)). *NeuMatC shares a conceptual parallel with PINNs in the use of structural knowledge to improve generalization. Specifically, PINNs incorporate physical laws (e.g., differential equations) to guide learning in scientific problems, while NeuMatC incorporates algebraic structures of matrix operations to guide learning.*

**Failure-Informed Adaptive Sampling.** In the loss function designed above, the way collocation points are selected is also important for enforcing the underlying algebraic structure. In this part, we introduce a ***failure-informed adaptive sampling strategy***, which adaptively identifies informative collocation points in regions where NeuMatC is more likely to violate the underlying structure.

This failure-informed adaptive sampling strategy is detailed as follows: At each iteration, the trained mapping $\mathbf{G}(\cdot)$ is evaluated on a set of candidate parameters, and we compute the residual $\mathscr{R}(\cdot)$ (see Section 2.2) to assess structural consistency. For each point $p$, we define a residual-based score $g(p) := \|\mathscr{R}(p)\|_F^2 - \epsilon_r$, where $\epsilon_r > 0$ is a predefined threshold. Points for which $g(p) > 0$ are considered to violate the algebraic structure identity and are thus labeled as failures. These points collectively form the *failure region* $\Omega_F = \{p \in \mathcal{P} \mid g(p) > 0\}$.

To quantify the significance of these violations, we estimate the proportion of candidate points that fall into the failure region, which we refer to as the *failure probability* $\hat{P}_{\mathcal{F}} = \frac{1}{N_s} \sum_{i=1}^{N_s} \mathbb{I}_{\Omega_F}(p_i)$, where $\mathbb{I}_{\Omega_F}$ is the indicator function and $\{p_i\}$ are uniformly sampled from the parameter domain.

If this estimated failure probability exceeds a given tolerance $\epsilon_p$, we select points from $\Omega_F$ with the largest residual scores $g(p)$ and add them to the collocation set for retraining. Otherwise, the refinement process terminates.

In our implementation, we adopt the widely used adaptive moment estimation (Adam) algorithm (Kingma & Ba, 2015). The effectiveness of adaptive sampling is further discussed in Section 4. ***The complete NeuMatC procedure is summarized in Algorithm 2.***

### 2.3 COMPLEXITY ANALYSIS

In this part, we present the complexity analysis of the proposed NeuMatC. Given a parametric matrix $\mathbf{A}(p) \in \mathbb{R}^{n_1 \times n_2}$, once NeuMatC is learned, operation results of $\mathbf{A}(p)$ for any $p \in \mathbb{P}$ are given by

$$\mathbf{G}_i(p) = \mathcal{C}_i \times_3 \Phi_{\theta_i}(p), \quad i = 1, \dots, m, \tag{4}$$

where $\Phi_{\theta_i} : \mathbb{R} \to \mathbb{R}^{d_i}$ is a MLP and $\mathcal{C}_i \in \mathbb{R}^{n_{1i} \times n_{2i} \times d_i}$ is a learned latent tensor. This computation involves only a few basic operations (i.e., matrix multiplications and nonlinear activations). Specifically, using $\mathbf{G}_i(p)$ as an example, the forward pass has complexity $\mathcal{O}(LW^2)$ for an $L$-layer network of width $W$ and the matrix-tensor product has complexity $\mathcal{O}(n_{1i}n_{2i}d)$. In contrast, conventional numerical methods for matrix operations, such as the Golub–Kahan algorithm for SVD (typically $\mathcal{O}(n_1 n_2^2)$ for $n_1 \geq n_2$ with large constant factors due to iterative procedures), require significantly higher computational cost and offer limited parallelism due to their sequential nature. ***Thus, by leveraging the low-rankness and continuity along the parameter dimension, NeuMatC offers a computationally advantageous alternative to conventional numerical solvers.***

## 3 EXPERIMENTAL RESULTS

In this section, we evaluate NeuMatC on representative tasks including matrix inversion and SVD on both synthetic and real-world parametric matrices. ***In many practical applications, achieving machine precision is often unnecessary (Dai et al., 2015). Our experiments show that NeuMatC offers significant speedup while maintaining acceptable accuracy.***

We also include an additional experiment in Appendix H to evaluate NeuMatC on large-scale parametric linear systems arising from PDE discretizations.

**Benchmarks.** We compare our proposed NeuMatC with a set of representative baselines on both matrix inversion and SVD tasks. For each task, we consider two categories: (i) ***pointwise methods***, which tackle matrix operations at different parameters independently without leveraging the relations along the parameter dimension; and (ii) ***parametric methods***, which exploit the continuous structure of matrices along the parameter dimension.

For matrix inversion, representative pointwise methods include: (1) *LU decomposition implemented in NumPy* (**LU (NumPy)**), a standard numerical algorithm applied independently at each parameter point; and (2) *Neural Newton Inversion* (**NNI**) (Zhang et al., 2008), which integrates neural networks with Newton-style iterations to approximate the inverse of a fixed matrix. Parametric methods include: (3) *Continuous Operator Interpolation* (**COI**) (Zahm & Nouy, 2016), which approximates inverse operators by interpolating a set of precomputed inverses along the parameter dimension; and (4) *ZNN-Inversion* (**ZNNI**) (Gerontitis et al., 2023), which extends ZNN to continuously compute matrix inverses using a neural dynamical system.

For matrix SVD, representative pointwise methods include: (1) *SVD implemented in NumPy* (**SVD (NumPy)**), which applies standard SVD independently at each parameter point; and (2) *Randomized SVD* (**RSVD**) (Halko et al., 2011), which computes approximate low-rank decompositions using random projections and is applied independently for each parameter. Parametric methods include: **(3) *Continuous Randomized SVD*** (**CRSVD**) (Kressner & Lam, 2024), which extends randomized SVD to parametric matrices by applying a shared random projection across the parameter dimension; and (4) *ZNN-SVD* (**ZNN-SVD**) (Chen & Zhang, 2020), which extends ZNN to continuously compute SVD using a neural dynamical system and employs the eighth-order discretization formula.

**Metrics.** To evaluate ***computational efficiency***, we report runtime and GFLOPs, *averaged over all $N_{test}$ test parameter points*. GFLOPs denote the total number of floating-point operations (in billions) required to produce the output, providing a hardware-agnostic measure of computational complexity. To assess ***computation quality***, we report the *Relative Error* (RelErr), also averaged over all test parameter points. Specifically, for matrix inversion we compute $\text{RelErr}_{\text{inv}} = \mathbb{E}_{p \in \mathcal{P}_{\text{test}}} \big[ \|\mathbf{A}(p)\hat{\mathbf{A}}^{-1}(p) - \mathbf{I}\|_F^2 / \|\mathbf{I}\|_F^2 \big]$, and for SVD we compute $\text{RelErr}_{\text{svd}} =$

$\mathbb{E}_{p \in \mathcal{P}_{\text{test}}}\left[\|\hat{\mathbf{U}}(p)\hat{\mathbf{S}}(p)\hat{\mathbf{V}}(p)^\top - \mathbf{A}(p)\|_F^2 / \|\mathbf{A}(p)\|_F^2\right]$. Here, $\mathcal{P}_{\text{test}}$ denotes the set of $N_{\text{test}}$ parameter points used for testing, and $\hat{\mathbf{A}}^{-1}(p)$, $\hat{\mathbf{U}}(p)$, $\hat{\mathbf{S}}(p)$, $\hat{\mathbf{V}}(p)$ are NeuMatC outputs.

*For reproducibility, we provide training details (including the experimental environment, sampling configuration, and hyperparameter choices) in Appendix F.* For operations such as SVD, where non-uniqueness may introduce discontinuities along the parameter dimension, we apply an alignment procedure on the training data following Tohidian et al. (2013) to enforce continuity across parameter samples. Detailed discussions on the alignment algorithm are provided in Appendix J.

### 3.1 SYNTHETIC DATA EXPERIMENTS.

**Setup.** In this part, we conduct experiments on 20 sets of synthetic data to evaluate the performance of the proposed NeuMatC. The parameterized matrices $\mathbf{H}(p) \in \mathbb{R}^{n \times n}$ are generated as $\mathbf{H}(p) = \mathbf{A}(p)\mathbf{B}(p)^\top + \varepsilon\mathbf{I}$, where $\mathbf{A}(p) = \mathbf{A}_0 \circ \sin(2\pi\mathbf{F}_A p + \mathbf{\Phi}_A)$ and $\mathbf{B}(p) = \mathbf{B}_0 \circ \cos(2\pi\mathbf{F}_B p + \mathbf{\Phi}_B)$, with both $\mathbf{A}(p), \mathbf{B}(p) \in \mathbb{R}^{n \times r}$. Here, $\circ$ denotes the elementwise product. The phase matrices $\mathbf{\Phi}_A, \mathbf{\Phi}_B$ are drawn uniformly from $[0, 2\pi]$, and the frequency matrices $\mathbf{F}_A, \mathbf{F}_B$ are drawn uniformly from $[0.5, 1.5]$. The coefficient matrices $\mathbf{A}_0, \mathbf{B}_0$ are sampled from standard normal distributions with column-wise energy decay. For NeuMatC, we uniformly sample $p$ from $[0, 1]$ at 40 parameter values for training and 100 for testing. Visualizations of the low-rank structure on one synthetic datasets are provided in Appendix F.

**Results.** Tables 1 and 2 show the results for both inversion and SVD tasks. Compared to pointwise methods, NNI and RSVD fall short in both accuracy and runtime due to underexploration of low-rankness and continuity. Compared to parametric methods, COI and ZNN-based baselines partially leverage parameter continuity but suffer from high runtime due to costly interpolation or sequential ODE integration. CRSVD improves efficiency via shared random projections but still incurs higher GFLOPs and lower accuracy. *In contrast, NeuMatC fully exploits both low-rankness and parameter continuity, achieving the best overall trade-off between accuracy and efficiency.*

Table 1: Matrix inversion performance at $n = 1024$. Results are averaged over 20 independently generated synthetic datasets. "MP" indicates machine precision.

| Method | CPU Time (ms) ↓ | GFLOPs↓ | RelErr ↓ |
|---|---|---|---|
| LU (NumPy) | $3.4 \times 10^1$ | $2.9 \times 10^0$ | MP |
| NNI | $2.7 \times 10^2$ | $2.6 \times 10^1$ | $9.2 \times 10^{-1} \pm 3.5 \times 10^{-1}$ |
| COI | $4.1 \times 10^3$ | $3.0 \times 10^0$ | $4.5 \times 10^{-3} \pm 4.6 \times 10^{-5}$ |
| ZNNI | $1.5 \times 10^3$ | $2.2 \times 10^2$ | $3.5 \times 10^{-3} \pm 9.3 \times 10^{-6}$ |
| NeuMatC | $\mathbf{4.3 \times 10^0}$ | $\mathbf{6.3 \times 10^{-2}}$ | $\mathbf{7.2 \times 10^{-4} \pm 1.8 \times 10^{-4}}$ |

Table 2: Matrix SVD performance at $n = 512$. Results are averaged over 20 independently generated synthetic datasets. "MP" indicates machine precision.

| Method | CPU Time (ms) ↓ | GFLOPs ↓ | RelErr ↓ |
|---|---|---|---|
| SVD (NumPy) | $6.1 \times 10^1$ | $1.1 \times 10^0$ | MP |
| RSVD | $1.2 \times 10^1$ | $4.2 \times 10^{-1}$ | $8.3 \times 10^{-3} \pm 2.3 \times 10^{-4}$ |
| CRSVD | $3.4 \times 10^0$ | $4.8 \times 10^{-1}$ | $1.5 \times 10^{-2} \pm 4.2 \times 10^{-3}$ |
| ZNN-SVD | $2.4 \times 10^3$ | $4.1 \times 10^0$ | $4.4 \times 10^{-1} \pm 2.4 \times 10^{-3}$ |
| NeuMatC | $\mathbf{9.8 \times 10^{-1}}$ | $\mathbf{7.5 \times 10^{-2}}$ | $\mathbf{6.4 \times 10^{-4} \pm 2.6 \times 10^{-4}}$ |

To better validate the low-rank structure exploited by NeuMatC, we conduct synthetic experiments where the low-rankness along the parameter dimension is explicitly controlled in Appendix I.

### 3.2 REAL DATA EXPERIMENTS.

**Setup.** Parametric matrix operations are widely encountered in wireless communication, where matrices such as MIMO channels vary continuously with parameters like carrier frequency. In these scenarios, repeated matrix inversion or SVD is often required across a large number of sampled states, posing significant computational challenges in latency-critical applications such as precoding, beamforming, and detection. To evaluate NeuMatC in a realistic scenario, we adopt the DeepMIMO dataset[1], a widely used simulation framework for millimeter-wave communication, to generate sequences of $256 \times 256$ real-valued MIMO channel matrices parameterized by carrier frequency [2]. Visualizations of the low-rank structure on the real datasets are provided in Appendix F.

[1] Available at `https://www.deepmimo.net/`.

[2] The DeepMIMO dataset gives complex parametric matrices. Without loss of generality, we extract the real part of the matrices for experiments. However, the proposed NeuMatC can be naturally extended to the complex case by separately modeling the real and imaginary components.

**Results.** Table 3 reports the results on real MIMO channel matrices for both inversion and SVD tasks. For matrix inversion, LU (NumPy) achieves machine precision but is relatively slow since it recomputes each inverse independently without leveraging parameter continuity. Among parametric baselines, COI achieves moderate accuracy but requires costly interpolation, while ZNNI partially exploits continuity but suffers from excessive runtime due to sequential ODE integration. For matrix SVD, SVD (NumPy) provides exact results but incurs higher runtime. RSVD and CRSVD accelerate computation via random projections, but their accuracy degrades, especially for CRSVD despite its shared projections across parameters. ZNN-SVD obtains relatively poor accuracy, and its runtime exceeds 200 ms per matrix. *In contrast, NeuMatC reduces runtime by about an order of magnitude compared with baselines while maintaining acceptable accuracy, highlighting NeuMatC's potential for real-world applications such as channel precoding and beamforming.*

Table 3: Performance comparison on real channel data for matrix inversion (left) and SVD (right). "MP" indicates machine precision.

| | (a) Matrix Inversion | | | (b) SVD | |
| --- | --- | --- | --- | --- | --- |
| **Method** | CPU Time (ms) ↓ | RelErr ↓ | **Method** | CPU Time (ms) ↓ | RelErr ↓ |
| **LU (NumPy)** | $1.8 \times 10^0$ | MP | **SVD (NumPy)** | $7.2 \times 10^0$ | MP |
| **NNI** | $2.7 \times 10^0$ | $6.5 \times 10^{-1}$ | **RSVD** | $1.6 \times 10^0$ | $8.7 \times 10^{-2}$ |
| **COI** | $2.4 \times 10^1$ | $2.3 \times 10^{-3}$ | **CRSVD** | $1.2 \times 10^0$ | $3.4 \times 10^{-1}$ |
| **ZNNI** | $5.3 \times 10^1$ | $1.4 \times 10^{-2}$ | **ZNN-SVD** | $2.0 \times 10^2$ | $1.3 \times 10^{-1}$ |
| **NeuMatC** | $\mathbf{5.8 \times 10^{-1}}$ | $\mathbf{8.3 \times 10^{-3}}$ | **NeuMatC** | $\mathbf{4.5 \times 10^{-1}}$ | $\mathbf{2.6 \times 10^{-3}}$ |

## 4 DISCUSSIONS

- *First, we further analyze the computational effectiveness of our NeuMatC.*

**Scalability with Matrix Size.** In this part, we examine how NeuMatC's computational efficiency scales with matrix size, using matrix inversion as a representative example. As shown in Figure 4 (a), conventional numerical solvers exhibit rapidly increasing time as matrix size grows. In contrast, NeuMatC maintains a nearly constant time across matrix sizes. This is because NeuMatC learns a parameter-to-result mapping, which requires only a few matrix multiplications and nonlinear activations that are highly parallelizable and hardware-efficient. These results demonstrate that NeuMatC is well suited for high-throughput applications such as wireless communication, where rapid and repeated matrix computations are required.

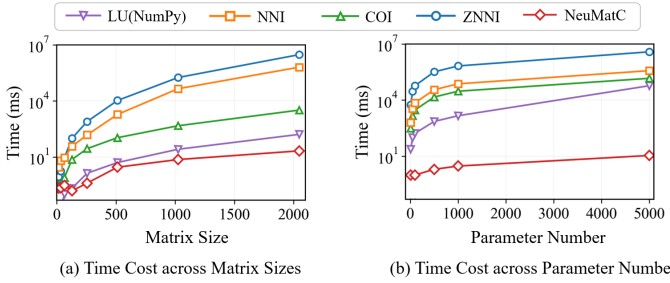

(a) Time Cost across Matrix Sizes          (b) Time Cost across Parameter Number

Figure 4: Time cost across (a) matrix size and (b) inferred parameter number on the inversion task. In subfigure (a), we report the average time cost for processing a single matrix, whereas in subfigure (b) we fix the matrix size at $512 \times 512$.

**Scalability with Inferred Parameter Number.** In many real-world scenarios such as wireless communications, it is often necessary to compute matrix operations at thousands of parameter points simultaneously. This motivates us to evaluate the scalability of NeuMatC with inferred parameter number. As shown in Figure 4(b), the runtime of baseline methods increases rapidly with the number of matrices, since these methods underutilize the continuity and low-rankness along the parameter dimension. In contrast, NeuMatC fully leverages this inherent structure. NeuMatC evaluates all parameter samples jointly through a single MLP forward pass followed by a matrix–tensor product, which involves only a small number of matrix multiplications and nonlinear operations and is highly optimized on modern hardware. As a result, the advantage of NeuMatC becomes increasingly significant as the number of matrices grows.

**Optimization on GPU Platforms.** Previous experiments were conducted on CPU to ensure fairness across baseline methods; however, NeuMatC can be further accelerated on GPU platforms. Its computations rely on matrix multiplications and nonlinear activations, which are highly optimized on GPUs. Table 4 reports NeuMatC's runtime across increasing matrix sizes on both CPU and GPU. Although NeuMatC already achieves low computation time on CPU, it benefits increasingly from GPU acceleration as the matrix size grows. These results confirm that NeuMatC can take effective advantage of GPU parallelism. Results of batched matrix operations on GPU platforms are demonstrated in Appendix I.

Table 4: Runtime and speedup of NeuMatC across larger matrix sizes on CPU and GPU.

| Matrix Size | 256 | 512 | 1024 | 2048 | 4096 |
|---|---|---|---|---|---|
| CPU Time (ms) | $2.6 \times 10^0$ | $7.8 \times 10^0$ | $4.1 \times 10^0$ | $1.6 \times 10^1$ | $6.3 \times 10^1$ |
| GPU Time (ms) | $2.4 \times 10^0$ | $3.3 \times 10^0$ | $6.0 \times 10^{-1}$ | $1.8 \times 10^0$ | $6.3 \times 10^0$ |
| Speedup | $1.1\times$ | $2.4\times$ | $6.8\times$ | $9.1\times$ | $10.1\times$ |

**Amortization of Training Cost.** In this part, we investigate the amortization benefit of NeuMatC's training cost by evaluating its performance on the SVD task. The datasets used in this experiment are generated following the same protocol described in our earlier synthetic data setup. We consider matrices of size $512 \times 512$ and vary the batch size from 512 to 2048. To prepare the supervised training data, we compute 40 SVDs by the cuSOLVER library. As shown in Table 5, we observe that NeuMatC still achieves substantial speedups over the high-performance GPU library cuSOLVER when the full runtime is considered. For example, when processing 2048 matrices, NeuMatC achieves nearly 30 times acceleration. These results justify NeuMatC's advantage in chanllenging applications such as wireless communication systems, which often involve computing over thousands of matrices Jeon et al. (2020); Gong et al. (2023).

Table 5: Overall runtime of NeuMatC (including data preparation, training, and inference) for SVDs of $512 \times 512$ matrices on the GPU platform. Results are averaged over 20 synthetic datasets. NeuMatC is trained for 500 epochs, reaching a relative error of approximately $1.0 \times 10^{-3}$.

| Batch size | cuSOLVER (ms) | NeuMatC (ms) | | | Speedup |
|---|---|---|---|---|---|
| | | Data Prep | Train | Inference | |
| 512 | $1.33 \times 10^4$ | $1.04 \times 10^3$ | $6.22 \times 10^3$ | 1.30 | $2.03\times$ |
| 1024 | $2.62 \times 10^4$ | $1.04 \times 10^3$ | $6.22 \times 10^3$ | 1.97 | $3.98\times$ |
| 2048 | $1.79 \times 10^5$ | $1.04 \times 10^3$ | $6.22 \times 10^3$ | 3.41 | $26.90\times$ |

- *Secondly, we further justify the generality of our NeuMatC.*

**Generalization on Multi-Dimensional Parameter Domains.** Although NeuMatC is introduced in the one-dimensional parameter case for clarity, the framework naturally generalizes to higher-dimensional parameter domains by extending $\Phi_\theta$ to take $p \in \mathbb{R}^d$ as input. To verify this capability, we conduct a two-dimensional inversion experiment on synthetic matrices of size $128 \times 128$. Each matrix is generated as $\mathbf{A}(p) = \mathbf{B}(p)\mathbf{B}(p)^\top + \varepsilon\mathbf{I}$, where $\mathbf{B}(p)$ is defined entries-wise by $\mathbf{B}_{i,j}(p_1, p_2) = \sum_k \alpha_{i,j,k}\,\phi_k(p_1, p_2)$ with $\{\phi_k\}$ being Fourier basis functions. Parameter values $(p_1, p_2)$ are sampled on a $50 \times 50$ grid. NeuMatC is trained using $5\%$ uniformly selected points and tested on 100 points. As shown in Table 6, NeuMatC maintains low relative error and fast inference, demonstrating its effectiveness in multi-dimensional case.

Table 6: NeuMatC on a 2D inversion task ($128 \times 128$). Results averaged over 20 synthetic datasets.

| CPU Time (ms) | RelErr |
|---|---|
| $3.7 \times 10^{-1}$ | $4.8 \times 10^{-4} \pm 2.4 \times 10^{-5}$ |

**Validation on Additional Matrix Operations.** NeuMatC is a general framework for learning parametric matrix operations and can naturally support a wide range of matrix operations. We have demonstrated its effectiveness on two representative tasks, i.e., matrix inversion and SVD. To further validate this generality, we apply NeuMatC to three additional operations, including QR decomposition, Cholesky decomposition, and matrix exponential. These experiments follow the same synthetic data generation protocol as in previous sections, using 20 independent datasets of parameterized $512 \times 512$ matrices for each task. For the Cholesky case, the matrices are constructed to be symmetric and positive definite, consistent with the requirements of the decomposition. As shown in Table 7, the framework maintains high accuracy and fast inference performance across these tasks.

Table 7: Performance of NeuMatC on different matrix computation tasks. Reported runtimes correspond to the average time to compute a single $512 \times 512$ matrix.

| Operation | RelErr | CPU Time (ms) | GPU Time (ms) |
|---|---|---|---|
| QR Decomposition | $4.1 \times 10^{-4} \pm 2.3 \times 10^{-4}$ | $2.3 \times 10^{0}$ | $4.5 \times 10^{-1}$ |
| Cholesky Decomposition | $6.8 \times 10^{-4} \pm 1.4 \times 10^{-4}$ | $7.9 \times 10^{-1}$ | $1.5 \times 10^{-1}$ |
| Matrix Exponential | $2.4 \times 10^{-3} \pm 2.6 \times 10^{-4}$ | $1.7 \times 10^{0}$ | $3.0 \times 10^{-1}$ |

- ***Thirdly, we conduct an ablation study on the NeuMatC framework.***

**Effectiveness of Adaptive Sampling.** We study the impact of the proposed failure-informed adaptive sampling strategy on the performance of NeuMatC. Specifically, we use the real-world SVD task introduced earlier as a representative case and fix the number of sampling points to 50. We compare two types of

Table 8: Influence of different sampling strategies on the real-world SVD task.

| Sampling | Random | Adaptive |
|---|---|---|
| RelErr $\downarrow$ | $1.2 \times 10^{-2}$ | $\mathbf{8.3 \times 10^{-3}}$ |

collocation point selection: **Random sampling**, where points are randomly sampled (starting with 50, adding 10 per round for 10 rounds), and **Adaptive sampling**, which follows the same schedule but uses the proposed strategy to select collocation points. As shown in Table 8, adaptive sampling achieves a lower RelErr, demonstrating its effectiveness in identifying informative collocation points and improving generalization and structural fidelity without requiring additional data.

**Analysis of NeuMatC Hyperparameters.** This part investigates how different hyperparameters impact the performance of NeuMatC. We consider the real-world matrix inversion task introduced in the previous experiments and vary key parameters, including the layer number $L$, layer width $W$, latent dimension $d$, frequency parameter $\omega$, balance weight $\lambda$, and the choice of activation function. As shown in Figure 5, NeuMatC exhibits robustness to hyperparameter variations: across a wide range of $L, W, d, \omega$, and $\lambda$, the RelErr consistently stays below $10^{-2}$, with only minor degradation at extreme settings.

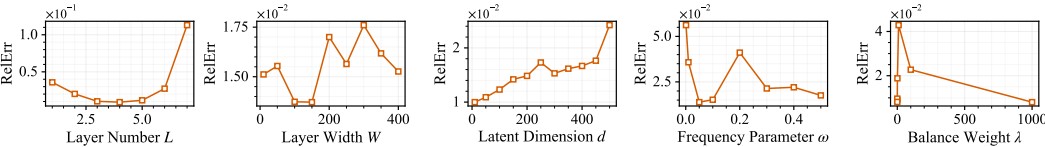

Figure 5: The quantitative performances of NeuMatC, w.r.t. different values of hyperparameters.

Table 9: Impact of activation function on NeuMatC.

| Activations | Sigmoid | GELU | ReLU | Tanh | Sine |
|---|---|---|---|---|---|
| RelErr ($\downarrow$) | $3.0 \times 10^{-2}$ | $1.5 \times 10^{-1}$ | $4.6 \times 10^{-2}$ | $2.6 \times 10^{-2}$ | $8.3 \times 10^{-3}$ |

The choice of activation function has a pronounced effect. As shown in Table 9, the sine activation obtains the lowest RelErr compared to classical activation functions such as ReLU, GELU, tanh, and sigmoid. This result aligns with prior findings on the benefits of periodic activation functions in representing fine-grained continuous signals (Sitzmann et al., 2020).

## 5 CONCLUSION

In this work, we propose NeuMatC, a general neural framework for efficient continuous parametric matrix operations. Unlike conventional numerical methods that tackle matrix operations at different parameter points independently, NeuMatC unsupervisedly learns a low-rank and continuous mapping from parameters to the results of matrix operations. Once trained, NeuMatC can directly map arbitrary parameters to their corresponding results using only a few basic operations (e.g., matrix multiplications and nonlinear activations), thereby significantly reducing redundant computational costs. Extensive experiments on both synthetic and real-world tasks demonstrate that NeuMatC achieves substantial speedups over classical methods while maintaining competitive accuracy.

We believe NeuMatC opens up a promising way for integrating AI into matrix computation, with strong potential for high-throughput processing of continuous parametric matrices commonly encountered in emerging real-world applications.

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

## A NOTATIONS

Table 10 presents the main notations used in our paper.

Table 10: The main notations used in the paper.

| Symbol | Description |
| --- | --- |
| $a \in \mathbb{R}$ | Scalar |
| $\mathbf{a} \in \mathbb{R}^n$ | Vector |
| $\mathbf{A} \in \mathbb{R}^{n_1 \times n_2}$ | Matrix |
| $\mathcal{A} \in \mathbb{R}^{n_1 \times n_2 \times n_3}$ | Tensor |
| $\mathbf{A}^\top$ | Transpose of matrix $\mathbf{A}$ |
| $\mathbf{A}^{-1}$ | Inverse of invertible matrix $\mathbf{A}$ |
| $\mathbf{I} \in \mathbb{R}^{n \times n}$ | Identity matrix |
| $\|\mathcal{A}\|_F$ | Frobenius norm of tensor: $\left(\sum_{i,j,k} |\mathcal{A}_{ijk}|^2\right)^{1/2}$ |
| $\|\mathcal{A}\|_1$ | $\ell_1$-norm of tensor: $\sum_{i,j,k} |\mathcal{A}_{ijk}|$ |

**Definition 1** (Mode-3 Tensor Folding and Unfolding (Kernfeld et al., 2015)). *The mode-3 unfolding of a tensor $\mathcal{A} \in \mathbb{R}^{n_1 \times n_2 \times n_3}$ is denoted by $\mathbf{A}_{(3)} \in \mathbb{R}^{n_3 \times n_1 n_2}$ and arranges the mode-3 fibers as the columns of $\mathbf{A}_{(3)}$. Concretely, the tensor element $\mathcal{A}_{(i_1, i_2, i_3)}$ maps to the matrix element $(\mathbf{A}_{(3)})_{(i_3, j)}$, where $j = i_1 + (i_2 - 1)n_1$. Mode-3 folding is the inverse operation of the mode-3 unfolding, represented by $\mathcal{X} = \text{Fold}_3(\mathbf{A}_{(3)})$.*

**Definition 2** (Mode-3 Matrix-Tensor Product (Kernfeld et al., 2015)). *The mode-3 matrix-tensor product of a third-order tensor $\mathcal{A} \in \mathbb{R}^{n_1 \times n_2 \times n_3}$ and a matrix $\mathbf{M} \in \mathbb{R}^{\hat{n}_3 \times n_3}$ is defined as*

$$\mathcal{A} \times_3 \mathbf{M} := \text{Fold}_3(\mathbf{M}\mathbf{A}_{(3)}),$$

*where $\mathcal{A} \times_3 \mathbf{M}$ is a tensor of size $n_1 \times n_2 \times \hat{n}_3$.*

# B   THEORETICAL JUSTIFICATION OF THE CONTINUITY OF PARAMETRIC MATRIX OPERATIONS

The proposed NeuMatC framework relies on the continuity of matrix operations with respect to the underlying parameter $p$. That is, given a parameter-dependent matrix $\mathbf{A}(p)$, the matrix operation output $\mathbf{G}(p) = \text{Operation}(\mathbf{A}(p))$ is expected to vary continuously with parameter $p$. This property ensures that the mapping $p \mapsto \mathbf{G}(p)$ can be efficiently learned. In this section, we present theoretical results that guarantee continuity for common matrix operations under mild assumptions [3] .

**Lemma 1** (Continuity of Matrix Product). *Let $\mathbf{A}(p) \in \mathbb{R}^{m \times \hat{n}}$ and $\mathbf{B}(p) \in \mathbb{R}^{\hat{n} \times n}$ be real analytic in $p \in \mathbb{P}$. Then the product $\mathbf{A}(p)\mathbf{B}(p) \in \mathbb{R}^{m \times n}$ is also real analytic in $p \in \mathbb{P}$.*

*Proof.* Each entry of $\mathbf{A}(p)\mathbf{B}(p)$ is given by

$$[\mathbf{A}(p)\mathbf{B}(p)]_{i,j} = \sum_{k=1}^{\hat{n}} \mathbf{A}_{i,k}(p) \cdot \mathbf{B}_{k,j}(p),$$

which is a finite sum of products of real analytic functions. Since the sum and product of analytic functions remain analytic, each entry of the result is analytic in $p$, hence $\mathbf{A}(p)\mathbf{B}(p)$ is analytic.   □

**Theorem 3** (Continuity of Matrix Inversion). *Let $\mathbf{A}(p) \in \mathbb{R}^{n \times n}$ be real analytic in $p \in \mathbb{P}$ and be nonsingular for all $p \in \mathbb{P}$. Then the inverse $\mathbf{A}^{-1}(p) \in \mathbb{R}^{n \times n}$ is also real analytic in $p$.*

*Proof.* The inverse can be written as

$$\mathbf{A}^{-1}(p) = \frac{1}{\det \mathbf{A}(p)} \cdot \text{adj}(\mathbf{A}(p)).$$

The determinant of $\mathbf{A}(p)$ is a finite sum of products of its entries and is therefore analytic. Under the assumption that $\det \mathbf{A}(p) \neq 0$ for all $p \in \mathbb{P}$, the reciprocal $\frac{1}{\det \mathbf{A}(p)}$ is analytic as well. Each entry of the adjugate matrix is a minor determinant, which is again an analytic function of $p$. Thus, the product is analytic, and so is $\mathbf{A}^{-1}(p)$.   □

**Theorem 4** (Continuity of Matrix SVD (Bunse-Gerstner et al., 1991)). *Let $\mathbf{A}(p) \in \mathbb{R}^{n_1 \times n_2}$ be real analytic in $p \in \mathbb{P}$. Then there exist real analytic matrices $\mathbf{U}(p) \in \mathbb{R}^{n_1 \times n_1}$, $\mathbf{S}(p) \in \mathbb{R}^{n_1 \times n_2}$, and $\mathbf{V}(p) \in \mathbb{R}^{n_2 \times n_2}$ such that*

$$\mathbf{A}(p) = \mathbf{U}(p)\mathbf{S}(p)\mathbf{V}(p)^\top, \quad \mathbf{U}(p)^\top \mathbf{U}(p) = \mathbf{I}_{n_1}, \quad \mathbf{V}(p)^\top \mathbf{V}(p) = \mathbf{I}_{n_2},$$

*where $\mathbf{S}(p)$ is diagonal for each $p$.*

The above results demonstrate that common matrix operations (e.g., matrix product, inversion, and SVD) preserve continuity along the parameter dimension under mild conditions.

**Remark 4** (Generality of Continuity Properties). *The continuity results above extend beyond inversion and SVD. Other operations, including eigenvalue decomposition, matrix exponentiation, QR decomposition, and pseudoinverse, also exhibit continuity under standard regularity assumptions. This suggests that the proposed framework is broadly applicable to a wide class of parametric matrix computation problems encountered in engineering and scientific applications.*

---

[3]To simplify the theoretical analysis, we consider parameter-dependent matrices whose entries are real analytic functions of the underlying parameter. Analyticity provides stronger regularity than continuity. In practice, many parametric matrices (such as those arising in wireless channel modeling, system transfer functions, and physics-based simulations) can be modeled by analytic forms.

## C  PROOF OF EXISTENCE OF THE COMPACT MAPPING (THEOREM 1)

***Proof of Theorem  1.*** For each $i$, consider the set of scalar-valued functions
$$\mathcal{F}_i := \{[\mathbf{G}_i(p)]_{jk} \mid 1 \le j \le n_{1i},\ 1 \le k \le n_{2i}\}.$$
By assumption, $\mathcal{F}_i$ spans a $d_i$-dimensional linear subspace.   Hence, there exists a basis $\{\varphi_{i,1}(p), \ldots, \varphi_{i,d_i}(p)\}$ such that for each pair $(j, k)$,
$$[\mathbf{G}_i(p)]_{jk} = \sum_{\ell=1}^{d_i} \mathcal{C}_{i,jk\ell}\, \varphi_{i,\ell}(p),$$
where $\mathcal{C}_{i,jk\ell}$ are fixed scalar coefficients. Collecting these coefficients over $(j, k, \ell)$ yields a tensor $\mathcal{C}_i \in \mathbb{R}^{n_{1i} \times n_{2i} \times d_i}$, and by defining $\Phi_i(p) := [\varphi_{i,1}(p), \cdots, \varphi_{i,d_i}(p)]^\top \in \mathbb{R}^{d_i}$ we obtain the compact representation
$$\mathbf{G}_i(p) = \mathcal{C}_i \times_3 \Phi_i(p).$$
Because the basis functions $\varphi_{i,\ell}$ exactly span the subspace, this identity holds for all $p \in \mathbb{P}$, which establishes the existence of the desired mapping $\widetilde{\mathbf{G}}_i(\cdot)$ and completes the proof.

## D  PROOF OF LIPSCHITZ CONTINUITY IN NEUMATC (THEOREM 2)

***Proof of Theorem  2.*** Given the mapping
$$\widetilde{\mathbf{G}}_i(\cdot) = \mathcal{C}_i \times_3 \Phi_{\theta_i}(\cdot), \tag{5}$$
the Frobenius norm of the difference between the outputs at $p_1$ and $p_2$ is given by
$$\begin{aligned}
\|\widetilde{\mathbf{G}}_i(p_1) - \widetilde{\mathbf{G}}_i(p_2)\|_F &= \left\| \mathcal{C}_i \times_3 \left( \Phi_{\theta_i}(p_1) - \Phi_{\theta_i}(p_2) \right) \right\|_F \\
&\le \|\mathcal{C}_i\|_1 \cdot \|\Phi_{\theta_i}(p_1) - \Phi_{\theta_i}(p_2)\|_1.
\end{aligned} \tag{6}$$
Using the assumption $\|\mathcal{C}_i\|_1 \le \kappa$, we have
$$\|\widetilde{\mathbf{G}}_i(p_1) - \widetilde{\mathbf{G}}_i(p_2)\|_F \le \kappa \cdot \|\Phi_{\theta_i}(p_1) - \Phi_{\theta_i}(p_2)\|_1. \tag{7}$$
Now we bound $\|\Phi_{\theta_i}(p_1) - \Phi_{\theta_i}(p_2)\|_1$. Let $\mathbf{h}^{(0)} = p$, and recursively define the MLP layers as
$$\mathbf{h}^{(\ell)} = \sigma(\mathbf{W}_\ell \mathbf{h}^{(\ell-1)}), \quad \text{for } \ell = 1, \ldots, d-1, \quad \Phi_{\theta_i}(p) = \mathbf{W}_d \mathbf{h}^{(d-1)}. \tag{8}$$
Since each activation is $L_\sigma$-Lipschitz and $\|\mathbf{W}_\ell\|_1 \le \eta$, we inductively obtain
$$\|\Phi_{\theta_i}(p_1) - \Phi_{\theta_i}(p_2)\|_1 \le (L_\sigma \eta)^d \cdot |p_1 - p_2|. \tag{9}$$
Combining the above results, we conclude:
$$\|\widetilde{\mathbf{G}}_i(p_1) - \widetilde{\mathbf{G}}_i(p_2)\|_F \le \kappa (L_\sigma \eta)^d \cdot |p_1 - p_2|. \tag{10}$$

## E  THEORETICALLY JUSTIFICATION OF THE LOW-RANK REPRESENTATION OF PARAMETRIC INVERSION

This section provides theoretical support for the low-rank representation admitted in the mapping $\mathbf{A}(p)^{-1} : \mathcal{P} \mapsto \mathbb{R}^{n \times n}$. Although
$$\mathbf{A}(p)^{-1} = \frac{1}{\det(\mathbf{A}(p))}\, \mathrm{adj}(\mathbf{A}(p))$$
is a rational function of the entries of $\mathbf{A}(p)$, $\mathbf{A}(p)^{-1}$ remains smooth on $p \in \mathcal{P}$ when $\mathbf{A}(p)$ stays invertible. Moreover, in many real-world applications (e.g., signal processing and wireless communication), $\mathbf{A}(p)$ commonly admits a low-rank representation along the parameter dimension. This combination of smoothness and low-rankness implies that $\mathbf{A}(p)^{-1} : \mathcal{P} \mapsto \mathbb{R}^{n \times n}$ admits an low-rank representation, which is theoretically justified by the following theorem.

**Theorem 5.** *Let $\mathcal{P} \subset \mathbb{R}$ be compact and $\mathbf{A} : \mathcal{P} \to \mathbb{R}^{n \times n}$ be invertible for all $p \in \mathcal{P}$. Assume that*

$$\mathbf{A}(p) = \mathcal{C}_{\mathbf{A}} \times_3 \psi(p), \qquad p \in \mathcal{P},$$

*where $\psi \in C^k(\mathcal{P}; \mathbb{R}^{r_{in}})$ and $\sigma_{\min}(\mathbf{A}(p)) \geq \delta > 0$ on $\mathcal{P}$. Then for any $\varepsilon > 0$, there exist a tensor $\mathcal{C}_{\mathbf{A}^{-1}} \in \mathbb{R}^{n \times n \times r_{out}}$ and a continuous function $\Phi : \mathcal{P} \to \mathbb{R}^{r_{out}}$ such that*

$$\sup_{p \in \mathcal{P}} \left\| \mathbf{A}(p)^{-1} - \mathcal{C}_{\mathbf{A}^{-1}} \times_3 \Phi(p) \right\| \leq \varepsilon,$$

*and the required rank satisfies*

$$r_{out} \ \leq \ C \, \varepsilon^{-\, r_{in}/k},$$

*where $C > 0$ is a constant independent of $\varepsilon$.*

*Proof.* Write $\psi(p) = (\psi_1(p), \ldots, \psi_{r_{in}}(p))$ and set $\Omega = \psi(\mathcal{P}) \subset \mathbb{R}^{r_{in}}$. Define the linear operator

$$\widehat{\mathbf{A}}(\mathbf{x}) := \mathcal{C}_{\mathbf{A}} \times_3 \mathbf{x} = \sum_{j=1}^{r_{in}} x_j \, \mathcal{C}_{\mathbf{A}}(:, :, j), \qquad \mathbf{x} \in \mathbb{R}^{r_{in}},$$

so that $\mathbf{A}(p) = \widehat{\mathbf{A}}(\psi(p))$ for every $p \in \mathcal{P}$. Since $\sigma_{\min}(\mathbf{A}(p)) \geq \delta$, we have

$$\sigma_{\min}(\widehat{\mathbf{A}}(\mathbf{x})) \geq \delta, \qquad \mathbf{x} \in \Omega,$$

hence each $\widehat{\mathbf{A}}(\mathbf{x})$ is invertible.

Because $\widehat{\mathbf{A}}$ is linear, it is a $C^\infty$ map. Matrix inversion is also $C^\infty$ on the Lie group

$$GL_n = \{\mathbf{M} \in \mathbb{R}^{n \times n} : \det \mathbf{M} \neq 0\}.$$

Since $\sigma_{\min}(\widehat{\mathbf{A}}(\mathbf{x})) \geq \delta$ for all $\mathbf{x} \in \Omega$, the matrices $\widehat{\mathbf{A}}(\mathbf{x})$ remain in a compact subset of $GL_n$, and hence the inverse mapping $\mathbf{x} \mapsto \widehat{\mathbf{A}}(\mathbf{x})^{-1}$ is $C^\infty$ on $\{\mathbf{x} : \sigma_{\min}(\widehat{\mathbf{A}}(\mathbf{x})) \geq \delta\}$. In particular, this inverse is of class $C^k$ on $\Omega$.

Let $Q \supset \Omega$ be a rectangle (i.e., a Cartesian product of bounded intervals) in $\mathbb{R}^{r_{in}}$. By classical multivariate Jackson–type approximation (applied entrywise to a $C^k$ extension of $\mathbf{x} \mapsto \widehat{\mathbf{A}}(\mathbf{x})^{-1}$ from $\Omega$ to $Q$), for any integer $K \geq 1$ there exists a matrix-valued polynomial of total degree at most $K$,

$$\mathbf{P}_K(\mathbf{x}) = \sum_{\substack{\alpha \in \mathbb{N}^{r_{in}} \\ |\alpha| = \alpha_1 + \cdots + \alpha_{r_{in}} \leq K}} \mathbf{B}_\alpha \, \mathbf{x}^\alpha, \qquad \mathbf{x}^\alpha = x_1^{\alpha_1} \cdots x_{r_{in}}^{\alpha_{r_{in}}}.$$

such that

$$\sup_{\mathbf{x} \in \Omega} \left\| \widehat{\mathbf{A}}(\mathbf{x})^{-1} - \mathbf{P}_K(\mathbf{x}) \right\| \leq C_1 K^{-k},$$

where $C_1 > 0$ depends only on $k, \delta, \mathcal{C}_{\mathbf{A}}$ and $\|\psi\|_{C^k}$. (Here $\|\psi\|_{C^k}$ denotes the usual $C^k$–norm of $\psi$ on $\mathcal{P}$.)

For $p \in \mathcal{P}$,

$$\mathbf{A}(p)^{-1} = \widehat{\mathbf{A}}(\psi(p))^{-1}, \qquad \mathbf{P}_K(\psi(p)) = \sum_{|\alpha| \leq K} \mathbf{B}_\alpha \, \psi(p)^\alpha,$$

and therefore

$$\sup_{p \in \mathcal{P}} \left\| \mathbf{A}(p)^{-1} - \mathbf{P}_K(\psi(p)) \right\| = \sup_{\mathbf{x} \in \Omega} \left\| \widehat{\mathbf{A}}(\mathbf{x})^{-1} - \mathbf{P}_K(\mathbf{x}) \right\| \leq C_1 K^{-k}.$$

Let $\{\alpha^{(1)}, \ldots, \alpha^{(d)}\}$ be the collection of all multi-indices $\alpha = (\alpha_1, \ldots, \alpha_{r_{in}}) \in \mathbb{N}^{r_{in}}$ with total degree $|\alpha| := \alpha_1 + \cdots + \alpha_{r_{in}} \leq K$, so that $r_{out} = \binom{K + r_{in}}{r_{in}}$. Define

$$\mathcal{C}_{\mathbf{A}^{-1}}(:, :, j) = \mathbf{B}_{\alpha^{(j)}}, \qquad \Phi_j(p) = \psi(p)^{\alpha^{(j)}},$$

where $\psi(p)^{\alpha^{(j)}} = \psi_1(p)^{\alpha_1^{(j)}} \cdots \psi_{r_{in}}(p)^{\alpha_{r_{in}}^{(j)}}$. Then, by the definition of the mode-3 tensor product,

$$\mathcal{C}_{\mathbf{A}^{-1}} \times_3 \Phi(p) = \sum_{j=1}^{d} \Phi_j(p) \, \mathcal{C}_{\mathbf{A}^{-1}}(:, :, j) = \mathbf{P}_K(\psi(p)).$$

Choose
$$K = \left\lceil (2C_1/\varepsilon)^{1/k} \right\rceil.$$

Then $C_1 K^{-k} \leq \varepsilon/2$, and after absorbing constants we obtain
$$\sup_{p\in\mathcal{P}} \left\| \mathbf{A}(p)^{-1} - \mathcal{C}_{\mathbf{A}^{-1}} \times_3 \Phi(p) \right\| \leq \varepsilon.$$

Finally, the rank estimate follows from
$$r_{\text{out}} = \binom{K + r_{\text{in}}}{r_{\text{in}}} \leq \frac{(K + r_{\text{in}})^{r_{\text{in}}}}{r_{\text{in}}!} \leq \frac{(2K)^{r_{\text{in}}}}{r_{\text{in}}!}, \qquad (K \geq r_{\text{in}}).$$

Since $K \leq (2C_1/\varepsilon)^{1/k} + 1$, we conclude that
$$r_{\text{out}} \leq C\,\varepsilon^{-\,r_{\text{in}}/k},$$

for a constant $C > 0$ depending only on $k$, $r_{\text{in}}$ and $C_1$, and independent of $\varepsilon$. $\qquad\square$

## F    EXPERIMENTAL SETTINGS

This section details the environment and configurations used in our experiments.

**Environment.** All experiments are conducted on a desktop with an NVIDIA RTX 3060 Ti GPU, Intel Core i7-12700KF CPU, and 32GB RAM. Our implementation uses Python and PyTorch 2.5.0 with CUDA 12.1.

**Sampling Configuration.** We perform all experiments in this normalized domain $[0, 1]$. For each experiment, we first generate a sampling set $\mathcal{S}_{\text{all}} = \{p_j\}_{j=1}^{N_s}$ by uniformly sampling $[0, 1]$, where $N_s$ is chosen from $\{20, 40, 60, 80\}$. The initial collocation set $\mathcal{D}_{\text{col}}^{(0)} = \{q_\ell^{(0)}\}_{\ell=1}^{N_c^{(0)}}$ is uniformly sampled from the remaining region $[0, 1] \setminus \mathcal{S}_{\text{all}}$, with $N_c^{(0)}$ selected from $\{20, 80, 140, 200\}$. During adaptive refinement, the collocation set is updated every $T$ iterations with $T \in \{100, 500, 1000\}$, and at each update $N_{\text{add}}$ points with the largest residual scores are added, where $N_{\text{add}}$ is chosen from $\{5, 10, 15, 20\}$.

**Experimental Configuration.** We follow the hyperparameter settings recommended in the original papers for all baseline methods. For NeuMatC, the number of hidden layers $L$ is selected from $\{2, 3, 4\}$ and the hidden layer width $W$ from $\{50, 100, 150, 200\}$. The activation function for all MLPs is $\sin(\omega\cdot)$, where the frequency parameter $\omega \in \{0.05, 0.10, 0.15, 0.20, 0.25\}$. The loss-balancing weight $\lambda$ is chosen from $\{0.1, 1, 10\}$, and the latent dimension $d$ from $\{10, 20, 30, 40\}$. For RSVD, the truncation rank of RSVD and CRSVD is set to 20 in synthetic data experiments and is set to 10 in real data experiment.

## G    LOW-RANK STRUCTURE IN THE USED PARAMETRIC DATASETS USED

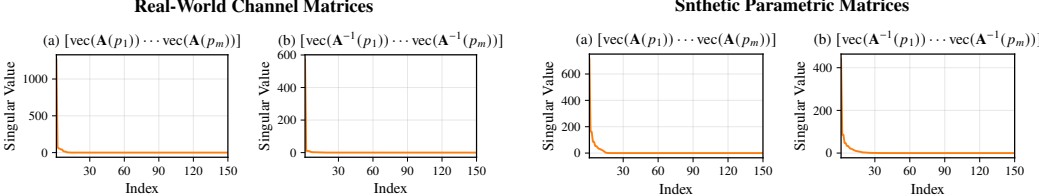

Figure 6: Singular value decay of matrices constructed by stacking vectorized forms of $\text{vec}(\mathbf{A}(p))$ and $\text{vec}(\mathbf{A}^{-1}(p))$ across $m$ parameter samples. Both real and synthetic datasets exhibit rapid singular value decay, revealing strong low-rank structure along the parameter dimension.

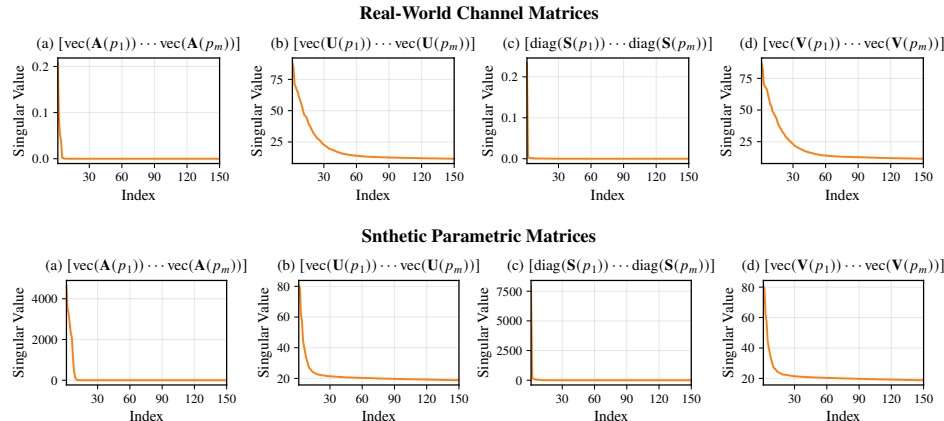

Figure 7: Singular value decay of matrices constructed by stacking vectorized $\text{vec}(\mathbf{H}(p))$ and vectorized SVD components $\text{vec}(\mathbf{U}(p))$, $\text{diag}(\mathbf{S}(p))$, and $\text{vec}(\mathbf{V}(p))$ across $m$ parameter samples. Both real-world and synthetic datasets exhibit rapid singular value decay, revealing strong low-rank structure along the parameter dimension.

# H  EXPERIMENTS ON SOLVING LARGE-SCALE PARAMETRIC LINEAR SYSTEMS ARISING FROM PDE DISCRETIZATION

In this part, we focus on solving large-scale parametric sparse linear systems, a fundamental problem in scientific computing and engineering applications involving parameterized PDEs. Such systems arise when discretizing PDEs with varying coefficients or boundary conditions across a parameter domain. They are challenging due to their high dimensionality and the need for efficient solutions across many parameter values. Traditional iterative methods like Krylov subspace recycling are commonly used but still require considerable computational effort for each instance. In the following, we present an experiment on a parametric advection–diffusion–reaction equation to demonstrate the applicability of NeuMatC to this class of problems, and compare its performance with a classical Krylov recycling solver (Parks et al., 2006).

**Advection-Diffusion-Reaction Equation.** We consider a one-parameter dependent advection–diffusion–reaction equation defined on the square domain $\Omega = [0,1]^2$ with homogeneous Dirichlet boundary conditions. The governing equation is

$$-\Delta u + \mathbf{v}(p) \cdot \nabla u + u = f, \tag{11}$$

where $z \in [0,1]$ is the parameter controlling the advection direction. The advection field is given by

$$\mathbf{v}(p) = D\cos(2\pi p)\,\mathbf{e}_1 + D\sin(2\pi p)\,\mathbf{e}_2,$$

with $D = 50$ and $\{\mathbf{e}_1, \mathbf{e}_2\}$ denoting the canonical basis vectors of $\mathbb{R}^2$.

A second-order finite difference discretization on a uniform mesh is employed, resulting in a system of equations of the form

$$\mathbf{A}(p)\,\mathbf{u}(p) = \mathbf{b},$$

where the system matrix $\mathbf{A}(p)$ depends continuously on $p$ and is assembled as

$$\mathbf{A}(p) = \mathbf{A}_0 + \cos(2\pi p)\,\mathbf{A}_1 + \sin(2\pi p)\,\mathbf{A}_2. \tag{12}$$

Here, $\mathbf{A}_0$, $\mathbf{A}_1$, and $\mathbf{A}_2$ are parameter-independent sparse matrices representing the discrete diffusion, advection, and reaction operators, respectively. The right-hand side vector $\mathbf{b}$ is independent of $z$ and corresponds to the discretized source term $f$.

**Experimental Setup** We consider solving a collection of 200 large-scale parametric sparse linear systems of size $10^5 \times 10^5$, generated from the advection–diffusion–reaction model described above. We compare NeuMatC with a Krylov subspace recycling method, which is widely used for efficiently solving sequences of related linear systems. We employ GCRO-DR (Parks et al., 2006) as a

representative method within this class, using an ILU(0) preconditioner and initializing each solve with the previous solution along the parameter dimension. All methods are implemented in Python and run on the same hardware platform. RelErr is measured as $\|\mathbf{A}(p)\mathbf{x}(p) - \mathbf{b}(p)\|_2/\|\mathbf{b}(p)\|_2$.

**Results**  Table 11 presents the runtime and accuracy comparisons between NeuMatC and the Krylov recycling baseline on solving 200 large-scale parametric sparse linear systems. NeuMatC achieves substantial inference speedup, reducing the total solution time from $5.19 \times 10^5$ ms to only 8.01 ms. When accounting for both the model training cost and the additional training data preparation cost (obtained by solving 20 systems using a sparse direct solver from the SciPy library), NeuMatC still achieves a competitive overall runtime. Compared to the Krylov recycling baseline, NeuMatC attains an overall $3.74\times$ speedup while maintaining comparable accuracy. These results highlight the strong potential of NeuMatC in large-scale sparse system problems.

Table 11: Comparison between Krylov subspace recycling and NeuMatC on parametric sparse linear systems arising from advection–diffusion–reaction PDEs. Each system has size $10^5 \times 10^5$, and inference times correspond to solving 200 systems.

| Method | CPU Time (ms) | | | RelErr | Speedup |
|---|---|---|---|---|---|
| | Data Preparation | Training | Inference | | |
| Krylov Recycling | — | — | $5.19 \times 10^5$ | $1.12 \times 10^{-3}$ | $1.00\times$ |
| NeuMatC | $1.2970 \times 10^4$ | $1.31 \times 10^5$ | $8.01$ | $1.11 \times 10^{-3}$ | $3.74\times$ |

## I   MORE RESULTS ON NUMERICAL EXPERIMENTS

**Results on Synthetic Datasets with Controlled Parametric Rank.**  To better validate the low-rank structure exploited by NeuMatC, we construct synthetic datasets where the low-rankness along the parameter dimension is explicitly controlled. Each matrix $\mathbf{H}(p)$ is constructed by simulating a parameter-dependent low-rank SVD:

$$\mathbf{H}(p) = \mathbf{U}(p)\,\mathbf{S}(p)\,\mathbf{V}(p)^\top,$$

where each factor (e.g., $\mathbf{U}(p)$) is represented as a mode-3 tensor product $\mathbf{U}(p) = \mathcal{C}_U \times_3 \boldsymbol{\phi}(p)$, with $\boldsymbol{\phi}(p) \in \mathbb{R}^d$ denoting smooth radial basis functions and $\mathcal{C}_U \in \mathbb{R}^{m \times r \times d}$ the coefficient tensor. The dimensions $d$ of these basis expansions determine the parametric rank. Similarly, the singular value matrix $\mathbf{S}(p)$ is constrained to be diagonal and constructed as $\mathbf{S}(p) = \mathcal{C}_S \times_3 \boldsymbol{\phi}(p)$.

We generate 200 parameterized matrices of size $512 \times 512$ for each test, varying the parametric rank $d \in \{5, 10, 15\}$ while fixing the SVD truncation rank $r = 100$. For NeuMatC, we uniformly sample $p$ from $[0, 1]$ at 20 parameter values for training and 100 for testing. Table 12 reports the performance of NeuMatC compared to randomized SVD across different parametric ranks. Compared to randomized SVD, NeuMatC consistently achieves both higher accuracy and faster inference, as randomized SVD relies on the stronger assumption that each individual matrix is low-rank.

Table 12: Comparison of NeuMatC and RSVD on synthetic $512 \times 512$ parametric matrices with controlled rank along the parameter dimension. The truncation rank of RSVD is set to $85$.

| Parametric Rank | NeuMatC | | RSVD | |
|---|---|---|---|---|
| | CPU Time (ms) | RelErr | CPU Time (ms) | RelErr |
| 10 | $1.45 \times 10^0$ | $3.90 \times 10^{-3}$ | $23.572 \times 10^1$ | $1.23 \times 10^{-2}$ |
| 20 | $3.81 \times 10^0$ | $2.70 \times 10^{-3}$ | $24.074 \times 10^1$ | $1.15 \times 10^{-2}$ |
| 30 | $3.77 \times 10^0$ | $4.61 \times 10^{-3}$ | $23.902 \times 10^1$ | $1.32 \times 10^{-2}$ |

**Results on Batched GPU Computation**  While GPU acceleration is evident in single-matrix inference, the benefits become even more pronounced in batched scenarios. In many real-world applications, such as wireless communication and signal processing, it is common to perform matrix

computations at hundreds or even thousands of parameter values simultaneously. These settings naturally form a batch of matrices, making parallel execution particularly important. To verify this advantage, we compare NeuMatC with cuSOLVER, a widely used batched high performance library on GPU, on batched inversion and SVD tasks. As shown in Table 13, NeuMatC consistently achieves significant speedups over cuSOLVER across various matrix and batch sizes, demonstrating its strong efficiency in batched GPU computation.

Table 13: Runtime comparison of NeuMatC and cuSOLVER for batched matrix inversion and SVD on GPU.

| Matrix Size | Batch Size | Matrix Inversion | | | SVD | | |
|---|---|---|---|---|---|---|---|
| | | cuSOLVER (ms) | NeuMatC (ms) | Speedup | cuSOLVER (ms) | NeuMatC (ms) | Speedup |
| | 128 | $2.7 \times 10^{-1}$ | $2.3 \times 10^{-1}$ | $1.17\times$ | $9.3 \times 10^{-1}$ | $5.3 \times 10^{-1}$ | $1.77\times$ |
| $32 \times 32$ | 256 | $2.9 \times 10^{-1}$ | $2.0 \times 10^{-1}$ | $1.49\times$ | $1.29 \times 10^{0}$ | $5.5 \times 10^{-1}$ | $2.35\times$ |
| | 512 | $4.2 \times 10^{-1}$ | $2.6 \times 10^{-1}$ | $1.62\times$ | $2.33 \times 10^{0}$ | $5.9 \times 10^{-1}$ | $3.98\times$ |
| | 128 | $2.47 \times 10^{1}$ | $7.8 \times 10^{-1}$ | $31.71\times$ | $3.31 \times 10^{3}$ | $1.44 \times 10^{0}$ | $2295.04\times$ |
| $512 \times 512$ | 256 | $5.48 \times 10^{1}$ | $1.22 \times 10^{0}$ | $44.87\times$ | $6.64 \times 10^{3}$ | $2.80 \times 10^{0}$ | $2374.41\times$ |
| | 512 | $1.15 \times 10^{2}$ | $2.28 \times 10^{0}$ | $50.36\times$ | $1.35 \times 10^{4}$ | $4.69 \times 10^{0}$ | $2882.23\times$ |

## J  CONTINUITY-AWARE DATA GENERATION STRATEGY

A key requirement for training NeuMatC is having ground-truth data that changes smoothly as the parameter varies. However, standard numerical solvers (like LAPACK's SVD) calculate each matrix $\mathbf{A}(p)$ separately. This independent processing causes two types of discontinuities that make it difficult for the network to learn a smooth mapping $\Phi_\theta$:

1. *Singular Value Swapping* : Standard solvers always sort singular values from largest to smallest ($\sigma_1 \geq \sigma_2 \dots$). When two singular value trajectories cross each other, the solver swaps their order to maintain the sorting. This causes the corresponding singular vectors to jump abruptly between indices.

2. *Sign/Phase Flipping*: Singular vectors $\mathbf{u}_i$ are not unique; they can be flipped by a negative sign ($s \in \{-1, 1\}$) or a complex phase ($e^{j\theta}$) without changing the validity of the SVD. Solvers often assign these signs randomly, leading to erratic flipping between steps.

To fix this, we use a continuity alignment procedure. We remove the strict sorting rule to allow smooth value changes and enforce phase consistency between adjacent parameter samples. Let the parameter domain be discretized as $p_0, \dots, p_N$. For the raw decomposition results at step $k$ $(\hat{\mathbf{U}}, \hat{\mathbf{\Sigma}}, \hat{\mathbf{V}})$, we search for a permutation matrix $\mathbf{P}$ (to fix the swapping) and a diagonal phase correction matrix $\mathbf{D}$ (to fix the flipping) to align them with the smooth results $\mathbf{U}_{k-1}$ from the previous step.

We formulate this as an optimization problem to maximize the similarity between the current and previous singular vectors:

$$\max_{\mathbf{P}, \mathbf{D}} \mathrm{Tr}\left(\mathrm{Re}\left(\mathbf{D}^H \mathbf{P}^T \underbrace{\hat{\mathbf{U}}^H \mathbf{U}_{k-1}}_{\mathbf{C}}\right)\right), \tag{13}$$

where $\mathbf{C}$ measures the correlation between the current raw vectors and the previous aligned vectors. $\mathrm{Re}(\cdot)$ takes the real part, and $\mathrm{Tr}(\cdot)$ (trace) sums up the similarity scores. Maximizing this objective effectively finds the best pairing and phase rotation to ensure the transition is as smooth as possible.

This optimization problem can be split into two steps:

**1. Optimal Phase Correction (D):** For any fixed permutation $\mathbf{P}$, let $\mathbf{M} = \mathbf{P}^T \mathbf{C}$ be the reordered correlation matrix. The trace term expands to the sum of the diagonal elements:

$$J = \sum_i \mathrm{Re}(d_i^* m_{ii}), \tag{14}$$

where $m_{ii}$ is the $i$-th diagonal element of $\mathbf{M}$, and $d_i$ is the $i$-th diagonal element of $\mathbf{D}$. Since $|d_i| = 1$, the term $\mathrm{Re}(d_i^* m_{ii})$ is maximized when the phase of $d_i$ cancels the phase of $m_{ii}$ (i.e., $d_i = m_{ii}/|m_{ii}|$). Under this optimal phase alignment, each term becomes the magnitude $|m_{ii}|$.

**2. Optimal Permutation (P):** Substituting the optimal phase back into the objective yields a simplified problem that depends only on $\mathbf{P}$:

$$\max_{\mathbf{P}} \sum_i |(\mathbf{P}^T \mathbf{C})_{ii}|. \tag{15}$$

Here, $(\mathbf{P}^T \mathbf{C})_{ii}$ represents the entry selected from the $i$-th column of $\mathbf{C}$ by the permutation. The problem reduces to finding a one-to-one mapping (permutation) that maximizes the sum of the absolute values of the selected entries. This is a standard *Linear Assignment Problem* (LAP) on the magnitude matrix $|\mathbf{C}|$.

While LAP can be solved globally using the Hungarian algorithm ($O(N^3)$), the continuity of the parameter space implies that $\mathbf{C}$ is strongly dominated by specific entries (i.e., vectors do not change arbitrarily between small steps). Therefore, a simple greedy selection strategy (picking the largest available element in each column) is both computationally efficient and sufficient for tracing smooth trajectories.

The detailed procedure is summarized in Algorithm 1.

---

**Algorithm 1** Continuity-Aware SVD Alignment Algorithm

---

**Require:** Sequence of parametric matrices $\{\mathbf{A}(p_k)\}_{k=0}^N$, standard SVD solver $\mathcal{S}(\cdot)$
**Ensure:** Continuous trajectories $\mathbf{U}(p_k), \boldsymbol{\Sigma}(p_k), \mathbf{V}(p_k)$
 1: **Initialize:** $[\mathbf{U}_0, \boldsymbol{\Sigma}_0, \mathbf{V}_0] \leftarrow \mathcal{S}(\mathbf{A}(p_0))$
 2: **for** $k = 1$ to $N$ **do**
 3:     **Step 1: Raw Decomposition**
 4:     $[\hat{\mathbf{U}}, \hat{\boldsymbol{\Sigma}}, \hat{\mathbf{V}}] \leftarrow \mathcal{S}(\mathbf{A}(p_k))$
 5:     **Step 2: Compute Correlation**
 6:     $\mathbf{C} \leftarrow \hat{\mathbf{U}}^H \mathbf{U}_{k-1}$
 7:     **Step 3: Solve for Permutation P**
 8:     Find permutation $\pi$ that maximizes $\sum_i |\mathbf{C}_{\pi(i),i}|$ via greedy search.
 9:     Construct permutation matrix $\mathbf{P}$ from $\pi$.
10:     Permute: $\tilde{\mathbf{U}} \leftarrow \hat{\mathbf{U}}\mathbf{P}, \quad \tilde{\mathbf{V}} \leftarrow \hat{\mathbf{V}}\mathbf{P}, \quad \boldsymbol{\Sigma}_k \leftarrow \mathbf{P}^T \hat{\boldsymbol{\Sigma}} \mathbf{P}$
11:     **Step 4: Solve for Phase D**
12:     Update correlation: $\tilde{\mathbf{C}} \leftarrow \mathbf{P}^T \mathbf{C}$
13:     Compute phases: $\mathbf{D} \leftarrow \mathrm{diag}(\mathrm{sign}(\mathrm{Re}(\tilde{\mathbf{C}}_{ii})))$
14:     Apply correction: $\mathbf{U}_k \leftarrow \tilde{\mathbf{U}}\mathbf{D}, \quad \mathbf{V}_k \leftarrow \tilde{\mathbf{V}}\mathbf{D}$
    **return** $\{\mathbf{U}_k, \boldsymbol{\Sigma}_k, \mathbf{V}_k\}_{k=0}^N$

---

## K  NEUMATC TRAINING PROCEDURE

The full NeuMatC procedure is outlined in Algorithm 2.

## L  LARGE LANGUAGE MODEL USAGE DISCLOSURE

Large language models (LLMs) were used solely to assist in improving the clarity of writing and identifying related literature. No part of the research ideation, methodology design, experimental setup, or analysis was performed by LLMs. All technical contributions were conceived and executed exclusively by the listed authors.

---

**Algorithm 2** NeuMatC

---

**Require:** Initial collocation set $\mathcal{D}_{\text{col}}^{(0)} = \{q_\ell^{(0)}\}_{\ell=1}^{N_c^{(0)}}$; supervised data $\mathcal{D}_{\text{data}} = \{(p_j, \mathbf{G}(p_j))\}_{j=1}^{N_s}$; maximum iterations $K_{\max}$; residual threshold $\epsilon_r$; failure probability tolerance $\epsilon_p$; update interval $T$; number of new collocation points $N_{\text{add}}$.

1:  Initialize parameters $\{\theta_i^{(0)}, \mathcal{C}_i^{(0)}\}_{i=1}^m$ with random values; set $k \leftarrow 0$
2:  **while** $k < K_{\max}$ **do**
3:      Optimize $\{\theta_i, \mathcal{C}_i\}_{i=1}^m$ via Adam to minimize:

$$\min_{\{\mathcal{C}_i,\theta_i\}_{i=1}^m} \mathcal{L}(\{\mathcal{C}_i, \theta_i\}; \mathcal{D}_{\text{data}}, \mathcal{D}_{\text{col}}^{(k)}) = \sum_{i=1}^m \sum_{j=1}^{N_s} \|\mathcal{C}_i \times_3 \Phi_{\theta_i}(p_j) - \mathbf{G}_i(p_j)\|_F^2$$
$$+ \lambda \sum_{q \in \mathcal{D}_{\text{col}}^{(k)}} \|\mathscr{R}(q; \{\mathcal{C}_i \times_3 \Phi_{\theta_i}(q)\}_{i=1}^m)\|_F^2.$$

4:      Let the mapping $\widehat{\mathbf{G}}^{(k)}(\cdot) := \{\widehat{\mathbf{G}}_i^{(k)}(\cdot)\}_{i=1}^m$, where each component is given by

$$\widehat{\mathbf{G}}_i^{(k)}(\cdot) := \mathcal{C}_i^{(k)} \times_3 \Phi_{\theta_i^{(k)}}(\cdot).$$

5:      **if** $k \bmod T = 0$ **then**
6:          Compute residual score: $g^{(k)}(p) = \|\mathscr{R}(p; \widehat{\mathbf{G}}^{(k)}(p))\|_F^2 - \epsilon_r$
7:          Estimate failure probability over candidate set $\mathcal{S}_{\text{cand}}$:

$$\hat{P}_{\mathcal{F}}^{(k)} = \frac{1}{|\mathcal{S}_{\text{cand}}|} \sum_{p \in \mathcal{S}_{\text{cand}}} \mathbb{I}\left[g^{(k)}(p) > 0\right]$$

8:          **if** $\hat{P}_{\mathcal{F}}^{(k)} < \epsilon_p$ **then break**
9:          Failure region: $\Omega_F^{(k)} = \{p \in \mathcal{S}_{\text{cand}} \mid g^{(k)}(p) > 0\}$
10:         Select $N_{\text{add}}$ points with largest $g^{(k)}(p)$: $\mathcal{P}_{\text{add}}^{(k)} \subset \Omega_F^{(k)}$
11:         Update collocation set: $\mathcal{D}_{\text{col}}^{(k+1)} = \mathcal{D}_{\text{col}}^{(k)} \cup \mathcal{P}_{\text{add}}^{(k)}$
12:     **else**
13:         $\mathcal{D}_{\text{col}}^{(k+1)} = \mathcal{D}_{\text{col}}^{(k)}$
14:     $k \leftarrow k + 1$
15: **return** trained mapping parameters $\{\theta_i^{(k)}, \mathcal{C}_i^{(k)}\}_{i=1}^m$

---

