# OpenReview forum: "NeuMatC: A General Neural Framework For Fast 	Parametric Matrix Operation"
_ICLR.cc/2026/Conference — Submitted to ICLR 2026_

### Official Review · Reviewer_4t5r · 2025-10-21

**Soundness:** 2
**Presentation:** 2
**Contribution:** 2
**Rating:** 4
**Confidence:** 4

**Summary:**

In this work, the authors propose a deep-learning-based method to efficiently perform operations on a one-parameter family of matrices.
The method is based on the assumption that the underlying structure of the transformed matrices is low-rank, i.e., that the matrix curve $p \mapsto G_i(p)$ spans a low-dimensional subspace of the space of matrices.

**Strengths:**

1. The problem the authors try to tackle is a very important one in the community of scientific computing.
2. The presentation is overall clear, the proposed method is easy, and it seems effective in terms of CPU timing.

**Weaknesses:**

1. For what concerns the problem of computing the SVD, there is a very large literature that studies the problem of doing it in an effective way, see e.g. [1]. The idea here is that if one has the first $A(0) = U(0)S(0)V(0)^\top$ factorized, then one can evolve the factors using $U(p),S(p),V(p)$ in an efficient way by integrating the equations
$$
\dot U(p) = (I-U(p)U(p)^\top) \dot A(p) V(p) S^{-1}(p), \quad \dot V(p) = (I-V(p)V(p)^\top) \dot A(p)^\top U(p) S^{-\top}(p), \quad \dot S(p) = U^\top(p) \dot A(p) V(p)
$$
Why didn't the authors compare with this family of methods, as these are known as state-of-the-art for parametric SVD computation, no one would ever compute the SVD pointwise.

2. I am a bit skeptical about the low-rankness assumption, especially in the case of parametric inversion. In particular, for inversion, the matrix $A(p)$ needs to be too, and in general, I fail to see why it is reasonable to assume that there is a low-rank structure hidden in the path $p \mapsto A(p)^{-1}$. Even more, it is known that the matrix inversion is a rational function of the entries (therefore not polynomial), therefore I would expect the path $p \mapsto A(p)^{-1}$ to not have low-rank "generically" in the paths [2].

3. As far as I understand, the rationale behind and the training itself resemble the idea of PINNS, as depicted in Figure 1, the NN takes as input the time parameter stamps and it outputs the matrix $G(p)$. Unfortunately, contrary to operator learning like methods, the proposed approach requires training a new model for each curve $p \mapsto A(p)$.

[1] O.Koch, C. Lubich, Dynamical low-rank approximation, SIMAX 2007.
[2] A. Pinkus,  Approximation theory of the MLP model in neural networks, Acta Numerica (1999), pp. 143-195

**Questions:**

1. Why in Tables 1,2,3 are the timings only in CPU? Could the authors also show the times on the GPU?
2. I understand the advantage of the proposed approach is at inference time,  but could the authors discuss a bit more in detail how expensive the model fitting?

I would also appreciate it if the authors could discuss the "weaknesses" section, as I believe fixing those points could significantly improve the quality of the work

---

> ### Author Response · Authors · 2025-11-22
> **Response to Reviewer 4t5r Part 1**
>
> We sincerely thank all reviewers for their positive comments (e.g., “quite novel”, “interesting and novel approach”, and “promising idea”) and their constructive suggestions. Our study addresses the emerging challenge of efficiently performing a large number of parametric matrix operations on moderate-sized matrices, which frequently arises in applications such as wireless communication and signal processing.
>
> To address this challenge, we propose **NeuMatC**, a general framework for accelerating parametric matrix operations. Specifically, NeuMatC learns an efficient mapping from parameters to matrix operation results by exploring the relations along the parameter dimension (i.e., inherent continuity and their relations in a low-rank structure). Numerical experiments demonstrate advantages over state-of-the-art baselines, achieving over $3\times$ speedup for parametric inversion and $10\times$ for parametric SVD in wireless communication tasks while maintaining good accuracy.
>
> For easy reference, a list of responses to the reviewers' comments has been compiled, and the corresponding changes have been highlighted in **blue** in the main text for ease of review. Should you need further information, please let us know. We look forward to hearing from you soon.

---

> ### Author Response · Authors · 2025-11-22
> **Response to Reviewer 4t5r Part 2**
>
> > **W1: For what concerns the problem of computing the SVD, there is a very large literature that studies the problem of doing it in an effective way, see e.g. [1]. The idea here is that if one has the first $A(0) = U(0)S(0)V(0)^{\top}$ factorized, then one can evolve the factors using $U(p), S(p), V(p)$ in an efficient way by integrating the equations
> > $$
> > \dot{U}(p) = (I - U(p)V(p)^{\top})\dot{A}(p)V(p)S^{-1}(p), \qquad
> > \dot{V}(p) = (I - V(p)V(p)^{\top})\dot{A}(p)^{\top}U(p)S^{-{\top}}(p), \qquad
> > \dot{S}(p) = U^{\top}(p)\dot{A}(p)V(p)
> > $$
> > Why didn't the authors compare with this family of methods, as these are known as state-of-the-art for parametric SVD computation, no one would ever compute the SVD pointwise?**
>
> **R1:** Thanks for the reviewer’s comment. In the introduction, we have discussed differential-equation-based approaches for parametric matrix computations, such as **ZNNI** [1] for matrix inversion and **ZNN-SVD** [2] for SVD. These methods evolve matrix factors (e.g., $U(p), S(p), V(p)$) at successive parameter values by integrating the corresponding equations.
> In the numerical experiments, we have included comparisons with **ZNNI** and **ZNN-SVD**. As shown in the tables below (i.e., Table 1 and Table 2 in the revised manuscript), NeuMatC achieves better accuracy while offering faster inference compared with ZNNI and ZNN-SVD. This is because such methods require costly integration at every parameter step and are highly sensitive to the smoothness of $A(p)$; if the data is not sufficiently smooth, large numerical errors arise.
>
> **Table: Comparison of NeuMatC with ZNN-I and ZNN-SVD on synthetic and real datasets for matrix inversion and SVD.**
>
> | Dataset   | Method   | CPU Time (ms) ↓   | RelErr ↓            | Method     | CPU Time (ms) ↓ | RelErr ↓            |
> |-----------|----------|--------------------|----------------------|------------|------------------|----------------------|
> | Synthetic | ZNNI     | 1.5 × 10³          | 3.5 × 10⁻³          | ZNN-SVD    | 2.4 × 10³        | 4.4 × 10⁻¹          |
> |           | NeuMatC  | **4.3 × 10⁰**      | **7.2 × 10⁻⁴**      | NeuMatC    | **9.8 × 10⁻¹**   | **6.4 × 10⁻⁴**       |
> | Real      | ZNNI     | 5.3 × 10¹          | 1.4 × 10⁻²          | ZNN-SVD    | 2.0 × 10²        | 1.3 × 10⁻¹          |
> |           | NeuMatC  | **5.8 × 10⁻¹**     | **8.3 × 10⁻³**      | NeuMatC    | **4.5 × 10⁻¹**   | **2.6 × 10⁻³**       |
>
>  **[1]** Dimitrios Gerontitis et al.   *A novel extended Li zeroing neural network for matrix inversion.*   Neural Computing and Applications, 35(19):14129–14152, 2023.
>
>  **[2]** Jianrong Chen and Yunong Zhang.   *Online singular value decomposition of time-varying matrices via zeroing neural dynamics.*   Neurocomputing, 383:314–323, 2020.

---

> ### Author Response · Authors · 2025-11-22
> **Response to Reviewer 4t5r Part 3**
>
> > **W2: I am a bit skeptical about the low-rankness assumption, especially in the case of parametric inversion. In particular, for inversion, the matrix $A(p)$ needs to be invertible too, and in general, I fail to see why it is reasonable to assume that there is a low-rank structure hidden in the path $p \mapsto A(p)^{-1}$. Even more, it is known that the matrix inversion is a rational function of the entries (therefore not polynomial), therefore I would expect the path $p \mapsto A(p)^{-1}$ to not have low-rank “generically” in the paths [2].**
>
> **R2:** Thanks for the insightful comment. We attempt to address your concern from both theoretical and numerical perspectives.
>
> **Theoretically**, although $A(p)^{-1} = \frac{1}{\det(A(p))}\operatorname{adj}(A(p))$ is a rational function of the entries of $A(p)$, $A(p)^{-1}$ remains smooth on $p \in \mathcal{P}$ when $A(p)$ stays invertible. Moreover, in many real-world applications (e.g., signal processing and wireless communication), $A(p)$ commonly admits a low-rank representation along the parameter dimension. This combination of smoothness and low-rankness implies that  $A(p)^{-1}: \mathcal{P} \mapsto \mathbb{R}^{n \times n}$  admits a low-rank representation, which is theoretically justified by the following theorem.
>
> >  **Theorem**  Let $\mathcal P \subset \mathbb{R}$ be compact and $\mathbf A:\mathcal P\to\mathbb R^{n\times n}$
> > continuous with all $\mathbf A(p)$ invertible. Assume
> >
> > $$ A(p)=\mathcal{C}_{A}\times_3\psi(p), \qquad p\in\mathcal{P}, $$
> >
> > where $\mathcal C_A\in\mathbb R^{n\times n\times r}$ and $\psi:\mathcal P\to\mathbb R^r$ is continuous.  Then for any $\varepsilon>0$, there exist $\mathcal C_{A^{-1}}\in\mathbb R^{n\times n\times d}$ and a continuous map $\Phi:\mathcal P\to\mathbb R^{d}$ such that
> >
> > $$ \sup_{p\in\mathcal P} \bigl\| A(p)^{-1} - \mathcal C_{A^{-1}}\times_3 \Phi(p)\bigr\| < \varepsilon ,$$
> >
> > where $d=\binom{K+r}{r}$ and $K\ge 0$ is an integer depending only on $\varepsilon$.   The detailed proof is given in Appendix E.
>
>
> **Numerically**, we provide empirical validation in a wireless communication setting, as shown in Fig. 2 of the manuscript. We sample parameter values $\{p_1, p_2, \ldots, p_m\}$ and form the matrix  $
> \begin{bmatrix}
> \operatorname{vec}(A(p_1)^{-1}) &
> \operatorname{vec}(A(p_2)^{-1}) &
> \cdots &
> \operatorname{vec}(A(p_m)^{-1})
> \end{bmatrix}
> \in \mathbb{R}^{n^2 \times m}.
> $  We observe that the matrix exhibits rapidly decaying singular values, indicating low-rankness along the parameter domain.

---

> ### Author Response · Authors · 2025-11-22
> **Response to Reviewer 4t5r Part 4**
>
> > **W3: As far as I understand, the rationale behind and the training itself resemble the idea of PINNs, as depicted in Figure 1, the NN takes as input the time parameter stamps and it outputs the matrix $\hat{G}(p)$. Unfortunately, contrary to operator learning–like methods, the proposed approach requires training a model for each curve $p \mapsto A(p)$.**
>
> **R3:** Thanks for the comment. Our work is motivated by practical needs in real-world applications such as wireless communication and signal processing~[3,4], where one must perform a large number of matrix operations over a parameterized matrix family. To meet this practical need, NeuMatC learns a mapping from the parameter to the corresponding matrix operation results, achieving efficient operations on the parameterized matrix family. Operator learning is a promising direction, and we would explore incorporating its philosophy into NeuMatC in future work.
>
> **[3]** Chanho Jeon, Zhenhao Li, and Christoph Studer.   *Approximate gram-matrix interpolation for wideband massive MU-MIMO systems.*  IEEE Transactions on Vehicular Technology, 69(5):4677–4688, 2020.
>
>  **[4]** Jiacheng Gong, Anirudh Kalia, and Minjie Yu.   *Scalable distributed massive MIMO baseband processing.*   In *Proceedings of the 20th USENIX Symposium on Networked Systems Design and Implementation (NSDI)*, pages 405–417, 2023.

---

> ### Author Response · Authors · 2025-11-22
> **Response to Reviewer 4t5r Part 5**
>
> > **Q1: Why in Tables 1,2 are the timings only in CPU? Could the authors also show the times on the GPU?**
>
> **R1:** We report CPU runtime and GFLOPs in Tables 1 and 2 to better reflect the inherent algorithmic complexity, as CPU measurements are less influenced by hardware-level optimizations than GPU timings.
>
> In the revised manuscript, we report the GPU runtime of NeuMatC and compare it with the high-performance GPU library cuSOLVER in the table below (i.e., Table 13 in the revised manuscript). We observe that NeuMatC achieves significant speedups for both matrix inversion and SVD across a wide range of matrix and batch sizes. These results demonstrate the potential of NeuMatC on modern GPU platforms.
>
> **Table: Runtime comparison of NeuMatC and cuSOLVER for batched matrix inversion and SVD on GPU.**
>
> | Matrix Size | Batch Size | cuSOLVER (ms) – Inversion | NeuMatC (ms) – Inversion | Speedup | cuSOLVER (ms) – SVD | NeuMatC (ms) – SVD | Speedup |
> |-------------|-------------|----------------------------|---------------------------|---------|----------------------|---------------------|---------|
> | 32×32 | 128 | 2.7 × 10⁻¹ | 2.3 × 10⁻¹ | 1.17× | 9.3 × 10⁻¹ | 5.3 × 10⁻¹ | 1.77× |
> |  | 256 | 2.9 × 10⁻¹ | 2.0 × 10⁻¹ | 1.49× | 1.29 × 10⁰ | 5.5 × 10⁻¹ | 2.35× |
> |  | 512 | 4.2 × 10⁻¹ | 2.6 × 10⁻¹ | 1.62× | 2.33 × 10⁰ | 5.9 × 10⁻¹ | 3.98× |
> | 512×512 | 128 | 2.47 × 10¹ | 7.8 × 10⁻¹ | 31.71× | 3.31 × 10³ | 1.44 × 10⁰ | 2295.04× |
> |  | 256 | 5.48 × 10¹ | 1.22 × 10⁰ | 44.87× | 6.64 × 10³ | 2.80 × 10⁰ | 2374.41× |
> |  | 512 | 1.15 × 10² | 2.28 × 10⁰ | 50.36× | 1.35 × 10⁴ | 4.69 × 10⁰ | 2882.23× |
>
>
> ---
>
> > **Q2: I understand the advantage of the proposed approach is at inference time, but could the authors discuss a bit more in detail how expensive the model fitting?**
>
> **R2:** Thanks for the helpful question. In the revised version, we report the overall runtime (including training and inference) of NeuMatC on the SVD task, as shown in the table below (i.e., Table 5 in the revised manuscript). We observe that NeuMatC still achieves substantial speedups over the high-performance GPU library cuSOLVER when the full runtime is considered. For example, when processing 2048 matrices, NeuMatC achieves nearly 30× acceleration. These results justify NeuMatC’s advantage in challenging applications such as wireless communication systems, which often involve computing over thousands of matrices [3,4].
>
> **Table: Overall runtime of NeuMatC (including training and inference) for SVDs of 512×512 matrices on the GPU platform.
> Results are averaged over 20 synthetic datasets. NeuMatC is trained for 500 epochs, reaching a relative error of approximately 1.0×10⁻³.**
>
> | Batch Size | cuSOLVER (ms) | NeuMatC Train (ms) | NeuMatC Inference (ms) | Speedup |
> |------------|----------------|---------------------|--------------------------|---------|
> | 512  | 1.33 × 10⁴ | 6.22 × 10³ | 1.30 | 2.15× |
> | 1024 | 2.62 × 10⁴ | 6.22 × 10³ | 1.97 | 4.22× |
> | 2048 | 1.79 × 10⁵ | 6.22 × 10³ | 3.41 | 28.75× |

---

> > ### Comment · Reviewer_4t5r · 2025-11-24
> >
> > First of all, I would like to thank the authors for their rebuttal.
> >
> > In order:
> >
> > **W1**. I still have some concerns left concerning the task of parametric SVD, as there is a really large body of literature of people attacking the same problem and producing stable and efficient integration algorithms exactly for this [1,2,3,4].
> > In particular, [3] is able to adapt the rank at integration time, and it solves stability problems observed in [1,4]. My concern is that the baselines provided by the authors do not represent the state of the art for this problem, and I think it would benefit in terms of quality of the work to include a comparison with one of the approaches based on Dynamical low-rank approximation (I would say [3] could probably be the best).
> >
> > **W2**. The authors are correct, I was just confused with the notion of "low-rank along the parameter dimension". I agree with the authors about the regularity of $A(p)^{-1}$ if the smallest singular value stays bounded away from zero. However, I think this definition of the low-rankness of a matrix curve could be misleading. As far as the proof goes, the authors seem to define a map $p \mapsto A(p)$ to have $\epsilon$-approximate rank $r$ if there exists $\mathcal C \in \mathbb R^{n \times n \times r}$ and $\Phi : \mathcal P \to \mathbb R^r$ such that
> > $$ ||A-\mathcal C \times_3 \Phi ||_{L^\infty(\mathcal P)} \leq \varepsilon.$$
> > If this is the definition of "low-rankness" along the parameter dimension, then I claim any continuous matrix curve $p \mapsto A(p)$ on a compact set $\mathcal P$ satisfies this property. In particular, this is a simple consequence of the Weierstrass theorem for continuous functions (as shown in the proof of the authors, in fact, the representation is nothing different than a polynomial expansion). In particular, Weierstrass theorem as it is, does not say anything about the relation between the regularity of the curve and the rank $r$, which a priori can grow even exponentially with the desired error $\epsilon$, making the definition of low-rank a bit too generic. I would appreciate hearing a comment from the authors on this point. Moreover, given this setting, why didn't the authors compare with a simple polynomial or rational fitting then? In this case, the map $\Phi$ would be fixed by the basis functions, and $\mathcal C$ would be the only set of parameters to fit.
> >
> > **W3**. Resolved by the authors.
> >
> > **Q1**. Resolved by the authors.
> >
> > [1] O.Koch, C. Lubich, Dynamical low-rank approximation, SIMAX 2007
> > [2] G. Ceruti, C. Lubich, An unconventional robust integrator for dynamical low-rank approximation, BIT Numerical Mathematics 2022.
> > [3] G. Ceruti, J. Kusch, C. Lubich, A rank-adaptive robust integrator for dynamical low-rank approximation, BIT Numerical Mathematics 2022.
> > [4] C. Lubich, I. Oseledets, A projector-splitting integrator for dynamical low-rank approximation, BIT Numerical Mathematics 2013.

---

> > > ### Author Response · Authors · 2025-11-28
> > > **Response to Reviewer 4t5r  Round 2 (Part 1)**
> > >
> > > We sincerely thank the reviewer for the valuable and constructive comments! We have carefully made more discussions and clarifications according to your comments.
> > >
> > > For easy reference, a list of responses to the reviewers' comments has been compiled, and the corresponding changes have been highlighted in blue in the main text for easy reference of our revision. Should you need further information, please let us know. We look forward to hearing from you soon.
> > >
> > > > **W1:**   I still have some concerns left concerning the task of parametric SVD, as there is a really large body of literature of people attacking the same problem and producing stable and efficient integration algorithms exactly for this [1,2,3,4]. In particular, [3] is able to adapt the rank at integration time, and it solves stability problems observed in [1,4].
> > > > My concern is that the baselines provided by the authors do not represent the state of the art for this problem, and I think it would benefit in terms of quality of the work to include a comparison with one of the approaches based on Dynamical low-rank approximation (I would say [3] could probably be the best).
> > > >
> > > > **[1]** O. Koch, C. Lubich, *Dynamical low-rank approximation*, SIMAX 2007.
> > > > **[2]** G. Ceruti, C. Lubich, *An unconventional robust integrator for dynamical low-rank approximation*, BIT Numerical Mathematics 2022.
> > > > **[3]** G. Ceruti, J. Kusch, C. Lubich, *A rank-adaptive robust integrator for dynamical low-rank approximation*, BIT Numerical Mathematics 2022.
> > > > **[4]** C. Lubich, I. Oseledets, *A projector-splitting integrator for dynamical low-rank approximation*, BIT Numerical Mathematics 2013.
> > >
> > > **Response:**
> > > Thanks for the comment. Dynamical low-rank approximation (DLA) methods approximate a parameter-dependent matrix by evolving its low-rank factors through numerical integration, which requires the existence of an explicit parametric representation. However, in many real-world scenarios such as wireless communications or massive MIMO systems, such parametric representations are not explicitly available. In contrast, NeuMatC is a general framework for accelerating parametric matrix operations and can perform tasks such as parametric SVD **without requiring an explicit representation**. NeuMatC can directly learn the mapping from parameters to matrix operation results using a few supervised samples, making NeuMatC applicable to a broad range of real-world problems. For the parametric SVD task, we compare NeuMatC with a representative DLA method [3]. As shown in the table below, NeuMatC achieves better accuracy and faster inference. This is because DLA methods require costly numerical integration at each step and are sensitive to the smoothness of the underlying parametric matrix.
> > >
> > > **Table: Comparison of NeuMatC with DLA on synthetic datasets for parametric SVD**
> > >
> > > | Methods | CPU Time (ms) | RelErr |
> > > |---------|----------------|---------|
> > > | DLA     | 7.1 × 10²      | 4.7 × 10¹ |
> > > | NeuMatC | **4.3 × 10⁰**  | **7.2 × 10⁻⁴** |

---

> > > ### Author Response · Authors · 2025-11-28
> > > **Response to Reviewer 4t5r Round 2 (Part 2)**
> > >
> > > > **W2:**   The authors are correct, I was just confused with the notion of “low-rank along the parameter dimension”.   I agree with the authors about the regularity of $A(p)^{-1}$ if the smallest singular value stays bounded away from zero.   However, I think this definition of the low-rankness of a matrix curve could be misleading.
> > >  As far as the proof goes, the authors seem to define a map $p \mapsto A(p)$ to have $\varepsilon$-approximate rank $r$ if there exist   $C \in \mathbb{R}^{n \times r}$ and $\Phi : \mathcal{P} \to \mathbb{R}^r$ such that
> > >  $$
> > >  \| A - C \times_3 \Phi \|_{L^\infty(\mathcal{P})} \le \varepsilon .
> > >  $$
> > >  If this is the definition of “low-rankness” along the parameter dimension, then I claim any continuous matrix curve   $p \mapsto A(p)$ on a compact set $\mathcal{P}$ satisfies this property.   In particular, this is a simple consequence of the Weierstrass theorem for continuous functions  (as shown in the proof of the authors, in fact, the representation is nothing different than a polynomial expansion).   In particular, Weierstrass theorem as it is does not say anything about the relation between the regularity of the curve and the rank $r$,   which a priori can grow even exponentially with the desired error $\varepsilon$, making the definition of low-rank a bit too generic.   I would appreciate hearing a comment from the authors on this point. Moreover, given this setting, why didn't the authors compare with a simple polynomial or rational fitting then?  In this case, the map $\Phi$ would be fixed by the basis functions, and $C$ would be the only set of parameters to fit.
> > >
> > > **Response:** Thank you for the insightful comments. In our work, a mapping  $\mathbf A:\mathcal P\to\mathbb R^{n\times n}$ is regarded as low-rank if there exist  $r$ (with $r \ll n^2$) basis matrices $\mathbf C_1,\dots,\mathbf C_r\in\mathbb R^{n\times n}$  such that $
> > > \mathbf A(p)\in\mathrm{span}\{\mathbf C_1,\dots,\mathbf C_r\},
> > > \ \text{for all } p\in\mathcal P. $
> > > In practice, the mapping is regarded as low-rank when a small number of basis matrices accounts for most of its total energy.
> > >
> > > In the revised manuscript, we further justify that the low-rankness and continuity of   $\mathbf A:\mathcal P \to \mathbb{R}^{n\times n}$  along the parameter dimension is inherited by its inverse  $\mathbf A(p)^{-1}:\mathcal P \to \mathbb{R}^{n\times n}$.  Specifically, we show that the parametric rank required to approximate  $\mathbf A(p)^{-1}$ grows only polynomially with the desired accuracy,  with the growth rate determined by the parametric rank $r_{\text{in}}$ and the  $k$-times differentiability of $\mathbf A:\mathcal P \to \mathbb{R}^{n\times n}$.
> > >
> > > >  **Theorem** Let $\mathcal P\subset\mathbb R$ be compact and   $\mathbf A:\mathcal P\to\mathbb R^{n\times n}$ be invertible for all $p\in\mathcal P$.   Assume that
> > >  $$\mathbf A(p) = \mathcal C_{\mathbf A}\times_3 \psi(p),\qquad p\in\mathcal P, $$
> > >  where $\psi\in C^k(\mathcal P;\mathbb R^{r_{\text{in}}})$ and   $\sigma_{\min}(\mathbf A(p))\ge\delta>0$ on $\mathcal P$.   Then for any $\varepsilon>0$, there exist a tensor   $\mathcal C_{\mathbf A^{-1}}\in\mathbb R^{n\times n\times {r_{\text{out}}}}$  and a continuous function   $\Phi:\mathcal P\to\mathbb R^{r_{\text{out}}}$ such that
> > >  $$ \sup_{p\in\mathcal P} \Bigl\| \mathbf A(p)^{-1} - \mathcal C_{\mathbf A^{-1}}\times_3 \Phi(p) \Bigr\|\le \varepsilon, $$
> > >  and the required rank satisfies
> > >  $${r_{\text{out}}} \le C \varepsilon^{-{r_{\text{in}}}/k}, $$
> > >  where $C>0$ is a constant independent of $\varepsilon$.
> > > > The detailed proof is given in Appendix E.
> > >
> > > NeuMatC provides a flexible framework that allows different choices of  $\Phi$, including polynomial functions, Fourier functions, rational functions, and neural  networks (MLPs). However, fixed functional bases are often unsuitable for capturing the complex  structure of matrix operations in practice, whereas neural networks  provide the expressive capacity needed. We take polynomial functions as an example and  conduct a comparison on real parametric MIMO channel matrices.  As shown in the table below, MLPs achieve lower  relative approximation error.
> > >
> > > **Table: Performance comparison of NeuMatC on the parametric SVD task with different function families.**
> > >
> > > | Class       | Polynomial          | Neural Networks      |
> > > |------------|----------------------|----------------------|
> > > | RelErr     | $8.7\times10^{-3}$   | $2.6\times10^{-3}$   |

---

### Official Review · Reviewer_SGgi · 2025-10-29

**Soundness:** 2
**Presentation:** 3
**Contribution:** 2
**Rating:** 4
**Confidence:** 3

**Summary:**

This paper addresses the problem of efficiently computing matrix operations such as matrix inversion or SVD for a family of matrices $A(p)$ parameterized by a variable $p \in \mathbb{R}$. To this end, the authors construct a neural network–based mapping with trainable parameters that directly maps the parameter $p$ to the result $G(p)$ of the matrix operation applied to $A(p)$. The design of this mapping relies on continuity and low-rankness assumptions on the function $p \mapsto G(p)$. The experimental results compare the proposed approach with traditional solvers, which cannot exploit such assumptions when computing a batch of matrix operations over a family of matrices $A(p)$ for multiple values of $p$. The authors show that these traditional solvers are slower than the proposed method, which can leverage these assumptions effectively.

**Strengths:**

* The experimental results on SVD seem to be sound. The method effectively shows a speedup compared to baselines without sacrificing accuracy.
* Theorem 1 motivates the design of a compact mapping, which makes the proposed method well suited for the SVD operation. Indeed, when the matrices \( A(p) \) are low-rank, the corresponding SVD components \( U(p) \), \( V(p) \), and \( S(p) \) are also low-rank, thereby satisfying the assumptions of Theorem 1.
* The authors proposed an ablation study to understand some key designs of the method: adaptive sampling, hyperparameters, etc.
* The proposed approach scales with the matrix sizes, and benefits from the parallelism in GPU because it relies only on GEMM or matrix-tensor product.

**Weaknesses:**

Theorem 1 might be misleading in some aspects when considering matrix inversion. The assumption of Theorem 1 is that the matrix operation results $G(p)$ are low-rank. However by definition an invertible matrix cannot be low-rank. From this point of view the Figure 2 might be misleading as well: it shows that both A(p) and A(p)^{-1} are close to low-rank, but this is not possible in general unless there are some assumption on A(p). Take for instance the matrices $H(p) = A(p) B(p)^\top + \epsilon I_n$ as considered in the experimental section 3.1 of the paper. By the Woodberry identity, we have that: $H(p)^{-1} = \frac{1}{\epsilon} I_n - \frac{1}{\epsilon^2} A(p) \Big(I_r + \tfrac{1}{\epsilon} B(p)^\top A(p)\Big)^{-1} B(p)^\top$. As $\epsilon \to 0$, $H(p)$ tends to a low-rank matrix, while its inverse behaves very differently: $H(p)^{-1} = \frac{1}{\epsilon}\Big(I - A(p)(B(p)^\top A(p))^{-1}B(p)^\top\Big) + o \left(\frac{1}{\epsilon}\right)$,
so $H(p)^{-1}$ becomes full rank and its norm grows like $1/\epsilon$. Therefore the application of the proposed method to matrix inversion needs a clarification from the authors.

**Questions:**

* In Tables 1 and 2, it is unclear whether the experimental protocol evaluates the proposed method on a **batch** of matrices or on a **single** matrix at a time. Since the primary application of the method concerns batched matrix operations, it is essential to compare its performance against batched versions of baseline algorithms (e.g., batched SVD in NumPy, batched randomized SVD, etc.). The authors should clarify this point to ensure a fair and meaningful validation of their results.
* To what extent can the proposed method be extended to other matrix operations, such as QR decomposition, LU decomposition, Cholesky decomposition, or interpolative decomposition?
* During training, how did the authors handle the non-uniqueness of the solutions to a given matrix operation? For example, the SVD decomposition is not unique when an eigenvalue has a multiplicity greater than one.

---

> ### Author Response · Authors · 2025-11-22
> **Response to Reviewer SGgi Part 1**
>
> We sincerely thank all reviewers for their positive comments (e.g., “quite novel”, “interesting and novel approach”, and “promising idea”) and their constructive suggestions. Our study addresses the emerging challenge of efficiently performing a large number of parametric matrix operations on moderate-sized matrices, which frequently arises in applications such as wireless communication and signal processing.
>
>    To address this challenge, we propose NeuMatC, a general framework for accelerating parametric matrix operations. Specifically, NeuMatC learns an efficient mapping from parameters to matrix operation results by exploring the relations along the parameter dimension (i.e., inherent continuity and their relations in a low-rank structure). Numerical experiments demonstrate advantages over state-of-the-art baselines, achieving over $3\times$ speedup for parametric inversion and $10\times$ for parametric SVD in wireless communication tasks while maintaining good accuracy.
>
>
>    For easy reference, a list of responses to the reviewers' comments has been compiled, and the corresponding changes have been highlighted in blue in the main text for easy reference of our revision. Should you need further information, please let us know. We look forward to hearing from you soon.

---

> ### Author Response · Authors · 2025-11-22
> **Response to Reviewer SGgi Part 2**
>
> > **W1: Theorem 1 might be misleading in some aspects when considering matrix inversion. The assumption of Theorem 1 is that the matrix operation results $G(p)$ are low-rank. However by definition an invertible matrix cannot be low-rank. From this point of view the Figure 2 might be misleading as well: it shows that both $A(p)$ and $A(p)^{-1}$ are close to low-rank, but this is not possible in general unless there are some assumption on $A(p)$.   Take for instance the matrices   $H(p) = A(p)B(p)^{\top} + \varepsilon I_n $ as considered in the experimental section 3.1 of the paper. By the Woodbury identity, we have   $ H(p)^{-1} = \tfrac{1}{\varepsilon} I_n - \tfrac{1}{\varepsilon^2} A(p) \left(I_r + \tfrac{1}{\varepsilon} B(p)^{\top} A(p)\right)^{-1} B(p)^{\top}. $
>  As $\varepsilon \to 0$, $H(p)$ tends to a low-rank matrix, while its inverse behaves very differently:  $H(p)^{-1} = \tfrac{1}{\varepsilon} \left( I - A(p)(B(p)^{\top}A(p))^{-1} B(p)^{\top} \right) + o\left( \tfrac{1}{\varepsilon} \right), $ so $H(p)^{-1}$ becomes full rank and its norm grows like $1/\varepsilon$. Therefore the application of the proposed method to matrix inversion needs a clarification from the authors.**
>
> **R1:** Sorry for the confusion. Our NeuMatC is designed as a general framework, and does not require the low-rankness assumptions of individual matrix $\mathbf{A}(p)$ or its inverse $\mathbf{A}^{-1}(p)$. Instead, we adopt a milder assumption commonly observed in real-world applications, i.e., **low-rankness along the parameter dimension**. That is, the collection $\{\mathbf{G}(p): p \in \mathcal{P}\}$ approximately lies in a low-dimensional subspace of $\mathbb{R}^{n \times n}$.
> For example, in Fig. 2, we compute singular values on the matrix
> $
> \begin{bmatrix}
> \operatorname{vec}(\mathbf{A}^{-1}(p_1)) & \cdots & \operatorname{vec}(\mathbf{A}^{-1}(p_m))
> \end{bmatrix},
> $
> where $\mathbf{A}(p)$ denotes a real-world channel matrix at parameter $p$ and $\{p_1, \dots, p_m\}$ are sampled parameter values. The rapid decay of singular values of this matrix indicates **low-rankness along the parameter dimension**.
> We have revised the manuscript to clarify this point.

---

> ### Author Response · Authors · 2025-11-22
> **Response to Reviewer SGgi Part 3**
>
> > **Q1: In Tables 1 and 2, it is unclear whether the experimental protocol evaluates the proposed method on a batch of matrices or on a single matrix at a time. Since the primary application of the method concerns batched matrix operations, it is essential to compare its performance against batched versions of baseline algorithms (e.g., batched SVD in NumPy, batched randomized SVD, etc.). The authors should clarify this point to ensure a fair and meaningful validation of their results.**
>
> **R1:** Thank you for the valuable comment. Tables 1 and 2 report the average performance over a batch of matrices, where each matrix is evaluated independently. We have clarified this point in the revised manuscript.
>
> In the discussion section, we have included comparisons with batched methods to further validate NeuMatC’s advantage. We compare NeuMatC against **cuSOLVER**, a widely used high-performance batched GPU library, on both matrix inversion and SVD tasks. The table below (Table 13 in the revised manuscript) shows that NeuMatC achieves substantial speedups over cuSOLVER, demonstrating the efficiency of NeuMatC in batched computation.
>
> **Table: Runtime comparison of NeuMatC and cuSOLVER for batched matrix inversion and SVD on GPU.**
>
> | Matrix Size | Batch Size | cuSOLVER (ms) – Inversion | NeuMatC (ms) – Inversion | Speedup | cuSOLVER (ms) – SVD | NeuMatC (ms) – SVD | Speedup |
> |-------------|-------------|----------------------------|---------------------------|---------|----------------------|---------------------|---------|
> | 32×32 | 128 | 2.7×10⁻¹ | 2.3×10⁻¹ | 1.17× | 9.3×10⁻¹ | 5.3×10⁻¹ | 1.77× |
> |  | 256 | 2.9×10⁻¹ | 2.0×10⁻¹ | 1.49× | 1.29×10⁰ | 5.5×10⁻¹ | 2.35× |
> |  | 512 | 4.2×10⁻¹ | 2.6×10⁻¹ | 1.62× | 2.33×10⁰ | 5.9×10⁻¹ | 3.98× |
> | 512×512 | 128 | 2.47×10¹ | 7.8×10⁻¹ | 31.71× | 3.31×10³ | 1.44×10⁰ | 2295.04× |
> |  | 256 | 5.48×10¹ | 1.22×10⁰ | 44.87× | 6.64×10³ | 2.80×10⁰ | 2374.41× |
> |  | 512 | 1.15×10² | 2.28×10⁰ | 50.36× | 1.35×10⁴ | 4.69×10⁰ | 2882.23× |

---

> ### Author Response · Authors · 2025-11-22
> **Response to Reviewer SGgi Part 4**
>
> > **Q2: To what extent can the proposed method be extended to other matrix operations, such as QR decomposition, LU decomposition, Cholesky decomposition, or iterative decomposition?**
>
> **R2:** NeuMatC is a flexible and general framework for learning parametric matrix operators and can be easily extended to a variety of matrix operations beyond inversion and SVD. In the revised manuscript, we demonstrate this generality by including results on QR decomposition, Cholesky decomposition, and matrix exponential. As shown in the table below (i.e., Table 7 in the revised manuscript), we observe that NeuMatC achieves consistently high accuracy and fast inference performance across these tasks.
>
> **Table: Performance of NeuMatC on different matrix computation tasks.
> Reported runtimes correspond to the average time to compute a single $512 \times 512$ matrix.**
>
> | Operation               | RelErr                                   | CPU Time (ms)      | GPU Time (ms)      |
> |------------------------|-------------------------------------------|---------------------|---------------------|
> | QR Decomposition       | 4.1 × 10⁻⁴ ± 2.3 × 10⁻⁴                   | 2.3 × 10⁰           | 4.5 × 10⁻¹          |
> | Cholesky Decomposition | 6.8 × 10⁻⁴ ± 1.4 × 10⁻⁴                   | 7.9 × 10⁻¹          | 1.5 × 10⁻¹          |
> | Matrix Exponential     | 2.4 × 10⁻³ ± 2.6 × 10⁻⁴                   | 1.7 × 10⁰           | 3.0 × 10⁻¹          |
>
>
> > **Q3: During training, how do the authors handle the non-uniqueness of the solutions to a given matrix operation? For example, the SVD decomposition is not unique when an eigenvalue has a multiplicity greater than one.**
>
> **R3:** NeuMatC is designed as a general framework for learning parametric matrix operators, and therefore does not introduce specific designs to explicitly handle non-uniqueness for generality. For operations such as SVD, where non-uniqueness may introduce discontinuities along the parameter dimension, we apply an alignment procedure on the training data following [1] to enforce continuity across parameter samples. We have made this detail more prominent in the revised manuscript.
>
>  **[1]** Mahdi Tohidian, Hamidreza Amindavar, and Ali M. Reza.   A DFT-based approximate eigenvalue and singular value decomposition of polynomial matrices.   *EURASIP Journal on Advances in Signal Processing*, 2013(93):1–11, 2013.

---

### Official Review · Reviewer_WYpt · 2025-10-30

**Soundness:** 2
**Presentation:** 3
**Contribution:** 1
**Rating:** 2
**Confidence:** 5

**Summary:**

The paper introduces NeuMatC, a neural framework designed to accelerate parametric matrix operations, such as inversion and singular value decomposition (SVD). The core idea is to leverage the assumed low-rank structure and continuity of the matrix operation results with respect to a varying parameter $p$. Instead of solving each matrix problem independently, NeuMatC learns a continuous mapping from the parameter $p$ to a low-dimensional latent representation, which is then used to reconstruct the final operation result (e.g., the inverse matrix or SVD components) via a multilayer perceptron (MLP) and a tensor product. The authors claim that, once trained, NeuMatC can achieve significant speedups (3-10x) over conventional methods like NumPy on both synthetic and real-world wireless communication datasets while maintaining acceptable accuracy.

**Strengths:**

**Significance:** The paper addresses the problem of parametric matrix computations, which is indeed significant and computationally demanding in many emerging applications like wireless communications, signal processing, and control systems. Developing methods to accelerate these repeated operations is a valuable research direction.

**Originality:** The core concept of learning a continuous, low-rank mapping from parameters directly to matrix operation results is an interesting and novel approach. It moves beyond treating each matrix instance independently and attempts to learn the underlying structure of the problem family, which is a promising idea.

 **Clarity:** The paper is generally well-written and clearly presents its proposed framework, NeuMatC. The motivation and the high-level mechanism of the algorithm are easy to follow.

**Weaknesses:**

#####

Despite the interesting premise, the paper suffers from significant weaknesses in its experimental evaluation, scope, and positioning within the existing literature. These issues are severe enough to undermine the paper's central claims.

1.  **Insufficient and Unfair Experimental Comparisons:** The experimental setup is not rigorous enough to support the claims of superiority.
    *   **CPU vs. GPU:** A major weakness is the comparison against inappropriate baselines. The proposed GPU-based method is compared primarily against CPU-based Python libraries (NumPy). This is not an apples-to-apples comparison. For a convincing evaluation, comparisons against GPU-accelerated libraries like **cuSOLVER** are essential.
    *   **Choice of Libraries:** High-performance scientific computing typically relies on highly optimized libraries in languages like C++, Fortran, or Julia (e.g., **SLEPc, Eigen, Arpack**), which are often significantly faster than their Python counterparts. The reported speedups against NumPy may be inflated due to the choice of a suboptimal baseline implementation.
    *   **Lack of Preconditioning:** The paper's comparisons with traditional iterative solvers appear to omit preconditioning. In practice, techniques like spectral transformation preconditioning are standard for solvers like LOBPCG and can dramatically accelerate convergence. Omitting them from the comparison makes the traditional methods appear much slower than they would be in a realistic setting and undermines the validity of the claimed speedup.
    *   **Neglect of Relevant Literature:** The paper fails to compare against a significant body of relevant work.
        *   There is no direct comparison with other neural network-based eigenvalue/SVD solvers that share a similar PINN-like architecture (e.g., **neuralSVD, PMNN, STNet** [1-3]). A comparison of accuracy and speed against these methods is necessary to position the contribution.
        *   More importantly, the paper ignores a rich field of traditional numerical methods designed specifically for series of related matrix problems. Techniques like **Krylov subspace recycling** [4], the **CHASE** algorithm [5], or using the invariant subspace from a previous solution as the initial guess for **LOBPCG** are designed to exploit the very same problem structure. A thorough comparison with these methods is essential to demonstrate the novelty and practical advantage of the proposed framework.

2.  **Limited Scope and Scalability Analysis:** The experiments are conducted in a narrow and relatively easy regime, which limits the generalizability of the findings.
    *   **Small Matrix Sizes:** The experiments are confined to relatively small matrices (dimension < 10,000). These problems can often be solved in seconds or less by traditional methods. The true challenge in industrial and scientific applications lies in large-scale matrices (e.g., dimensions from $10^4$ to $10^6$). The paper fails to demonstrate the method's performance in this critical regime. Comparisons should be made against state-of-the-art iterative solvers like **Krylov-Schur, LOBPCG, and JD** (e.g., using implementations from SLEPc [6]) for such large-scale problems.
    *   **Limited Matrix Types:** The evaluation should consider both **sparse matrices** (common in PDE discretizations) and **dense matrices** (e.g., from quantum chemistry). The paper's focus on low-rank matrices from communication is a significant limitation. The performance on non-low-rank matrices, such as those arising from PDE stability analysis (e.g., modal analysis of Maxwell's equations), is not explored. If the method is only applicable to low-rank problems, this limitation must be clearly stated in the abstract and introduction.

3.  **Inadequate Justification of Low-Rank Claims:** While the paper is motivated by the low-rank property of parametric matrices, the experimental validation is superficial.
    *   The authors do not provide details on the rank structure of their test matrices (e.g., a plot of singular value decay or loss versus rank). This makes it difficult to assess how "low-rank" the problems truly are and how well the method exploits this property.
    *   To properly validate the low-rank performance, the authors should construct synthetic datasets with explicitly controlled ranks and compare against specialized low-rank SVD algorithms. Simply testing on real-world datasets is insufficient.
    *   When comparing to randomized SVD, crucial details like the choice of the compression dimension (rank) are omitted. The performance of randomized SVD is highly dependent on this parameter, which should be chosen based on the target rank or error tolerance [7].

[1] Operator SVD with neural networks via nested low-rank approximation, ICML 2024

[2] Neural networks based on power method and inverse power method for solving linear eigenvalue problems, Computers & Mathematics with Applications 2023

[3] STNet: Spectral Transformation Network for Solving Operator Eigenvalue Problem,  NeurIPS 2025

[4] Recycling Krylov subspaces for sequences of linear systems, SIAM 2006

[5] ChASE: Chebyshev accelerated subspace iteration eigensolver for sequences of Hermitian eigenvalue problems, ACM 2019

[6] SLEPc: A scalable and flexible toolkit for the solution of eigenvalue problems, ACM 2015

[7] Finding Structure with Randomness: Probabilistic Algorithms for Constructing Approximate Matrix Decompositions, SIAM 2011

**Questions:**

My decision could potentially be changed if the authors can provide satisfactory answers to the following fundamental questions:

1.  **Handling of Singular Value/Vector Crossings:** A critical concern is the handling of singular value/vector crossings. As a parameter $p$ varies continuously, the ordering of singular values can change (e.g., the first and second singular values swap). A neural network trained to output the "k-th" singular vector would likely fail at such crossing points, as it would need to abruptly switch its output from one vector function to another, which is a discontinuous behavior. How does the proposed framework handle this well-known phenomenon in parametric eigenvalue problems? Were the datasets specifically chosen to avoid such crossings? A discussion of this issue can be found in literature such as [1] P376.

2.  **Scalability and Fair Baselines:** Could the authors provide performance benchmarks on truly large-scale matrices (e.g., dimensions from $10^4$ to $10^6$), for both sparse and dense cases? In this context, could you compare against established large-scale iterative solvers like **Krylov-Schur, LOBPCG, and JD** (e.g., from SLEPc), including fair comparisons on GPU (e.g., vs. **cuSOLVER**) and with appropriate **preconditioning**?

3.  **Practical Applicability and Overhead:** The proposed method requires a pre-generated training set of related matrices.
    *   In a real-world scenario where matrix problems may arise independently and without a unified parametric representation, what is the applicable scope of this method?
    *   When comparing against traditional solvers that work on a single instance, should the overhead of data generation and network training be included in the timing? What are the specific application scenarios where generating such a large, correlated dataset is feasible and the amortization of training cost makes this approach superior to traditional methods that can exploit problem-to-problem correlation without pre-training (e.g., **Krylov subspace recycling** [1], the **CHASE** algorithm [2], or using the invariant subspace from a previous solution as the initial guess for **LOBPCG**)?

4.  **Low-Rank Analysis:** To substantiate the claims about leveraging low-rank structure, could the authors provide a more detailed analysis? This should include: (a) visualizations of the singular value decay for the matrices in your datasets, (b) an ablation study showing performance on synthetic datasets with explicitly controlled ranks, and (c) the specific configuration details for the randomized SVD baseline, particularly the rank of the projection.

[1] Spectra and Pseudospectra by Trefethen and Embree

[2] Recycling Krylov subspaces for sequences of linear systems, SIAM 2006

[3] ChASE: Chebyshev accelerated subspace iteration eigensolver for sequences of Hermitian eigenvalue problems, ACM 2019

---

> ### Author Response · Authors · 2025-11-22
> **Response to Reviewer WYpt Part 1**
>
> We sincerely thank all reviewers for their positive comments (e.g., “quite novel”, “interesting and novel approach”, and “promising idea”) and their constructive suggestions.
>
> We appreciate your expertise in classical numerical linear algebra, particularly in large-scale sparse problem solving. However, our work focuses on a different and emerging challenge arising from real-world applications (e.g., 5G/6G wireless communication): efficiently performing large numbers of parametric matrix operations on moderate-sized matrices.
>
> To address this challenge, we propose NeuMatC, a general framework for accelerating parametric matrix operations. Specifically, NeuMatC learns an efficient mapping from parameters to matrix operation results by exploring the relations along the parameter dimension (i.e., inherent continuity and their relations in a low-rank structure). Numerical experiments demonstrate the advantages over state-of-the-art baselines.
>
> For easy reference, a list of responses to the reviewers' comments has been compiled, and the corresponding changes have been highlighted in **blue** in the main text for easy reference of our revision. Should you need further information, please let us know. We look forward to hearing from you soon.

---

> ### Author Response · Authors · 2025-11-22
> **Response to Reviewer WYpt Part 2**
>
> > **W1** Insufficient and Unfair Experimental Comparisons: The experimental setup is not rigorous enough to support the claims of superiority.
>
> > **W1.1**  (CPU vs. GPU) A major weakness is the comparison against inappropriate baselines. The proposed GPU-based method is compared primarily against CPU-based Python libraries (NumPy). This is not an apples-to-apples comparison. For a convincing evaluation, comparisons against GPU-accelerated libraries like **cuSOLVER** are essential.
>
> **R1.1:** Thank you for the comment. NeuMatC is not GPU-based; it is a general framework whose learned model can run on both CPU and GPU. In Tables 1–2, we report CPU runtime and GFLOPs to better reflect the intrinsic algorithmic complexity of each method, since CPU measurements are less affected by hardware-dependent kernel optimizations than GPU timings. This allows a cleaner apples-to-apples comparison of the core algorithms.
>
> In the revised manuscript, we have included comparisons with GPU-accelerated solvers (e.g., cuSOLVER). From the table below (i.e., Table 13 in the revised manuscript), we observe that NeuMatC achieves substantial speedups over cuSOLVER on both inversion and SVD tasks, demonstrating the efficiency of NeuMatC on GPU platforms.
>
> **Table: Runtime comparison of NeuMatC and cuSOLVER for batched matrix inversion and SVD on GPU.**
>
> | Matrix Size | Batch Size | cuSOLVER (ms) – Inversion | NeuMatC (ms) – Inversion | Speedup | cuSOLVER (ms) – SVD | NeuMatC (ms) – SVD | Speedup |
> |-------------|------------|----------------------------|---------------------------|---------|----------------------|---------------------|---------|
> | 32×32 | 128 | 2.7 × 10⁻¹ | 2.3 × 10⁻¹ | 1.17× | 9.3 × 10⁻¹ | 5.3 × 10⁻¹ | 1.77× |
> |  | 256 | 2.9 × 10⁻¹ | 2.0 × 10⁻¹ | 1.49× | 1.29 × 10⁰ | 5.5 × 10⁻¹ | 2.35× |
> |  | 512 | 4.2 × 10⁻¹ | 2.6 × 10⁻¹ | 1.62× | 2.33 × 10⁰ | 5.9 × 10⁻¹ | 3.98× |
> | 512×512 | 128 | 2.47 × 10¹ | 7.8 × 10⁻¹ | 31.71× | 3.31 × 10³ | 1.44 × 10⁰ | 2295.04× |
> |  | 256 | 5.48 × 10¹ | 1.22 × 10⁰ | 44.87× | 6.64 × 10³ | 2.80 × 10⁰ | 2374.41× |
> |  | 512 | 1.15 × 10² | 2.28 × 10⁰ | 50.36× | 1.35 × 10⁴ | 4.69 × 10⁰ | 2882.23× |
>
>
> ---
>
> > **W1.2 (Choice of Libraries):**
> > High-performance scientific computing typically relies on highly optimized libraries in languages like C++, Fortran, or Julia (e.g., **SLEPc**, **Eigen**, **Arpack**), which are often significantly faster than their Python counterparts. The reported speedups against NumPy may be inflated due to the weakness of a suboptimal baseline implementation.
>
> **R1.2:** We compare NeuMatC against the highly optimized C++ library `Eigen`. As shown in the table below, NeuMatC achieves significantly faster inference compared to Eigen's SVD implementation. This is attributed to NeuMatC relying only on a few basic operations (e.g., matrix multiplications and nonlinear activations), whereas traditional libraries like Eigen involve a larger number of computationally intensive operations.
>
> **Table: Comparison of single-instance runtime (512×512 matrices) between NeuMatC (C++ forward inference) and Eigen (C++ SVD).**
>
> | Method | NeuMatC (C++) | Eigen (C++) |
> |--------|----------------|-------------|
> | Inference Time (ms) | 4.87 × 10⁰ | 2.79 × 10³ |
>
>
> ---
>
> > **W1.3 (Lack of Preconditioning):**
> > The paper’s comparisons with traditional iterative solvers appear to omit preconditioning. In practice, techniques like spectral transformation preconditioning are standard for solvers such as **LOBPCG** and can dramatically accelerate convergence. Omitting them from the comparison makes the traditional methods appear much slower than they would be in a realistic setting and undermines the validity of the claimed speedup.
>
> **R1.3:** Thanks for the comment. The traditional solvers used in our experiments are direct solvers from high-performance libraries such as NumPy, rather than iterative eigensolvers like LOBPCG. Therefore, preconditioning is not applicable in these comparisons.

---

> > ### Author Response · Authors · 2025-11-22
> > **Response to Reviewer WYpt Part 8**
> >
> > > **Q4 (Low-Rank Analysis):**
> > > To substantiate the claims about leveraging low-rank structure, could the authors provide a more detailed analysis? This should include:
> > > (a) visualizations of the singular value decay for the matrices in your datasets,
> > > (b) an ablation study showing performance on synthetic datasets with explicitly controlled ranks, and
> > > (c) the specific configuration details for the randomized SVD baseline, particularly the rank of the projection.
> >
> > **R4:** Thank you for the comment. Traditional low-rank methods such as randomized SVD typically rely on a strong assumption that each individual matrix is low-rank. In contrast, NeuMatC does not assume low-rankness of the individual matrices. We adopt a *milder* assumption commonly observed in real-world applications, i.e., **low-rankness along the parameter dimension**, where the collection  $
> > \{G(p): p \in \mathcal{P}\}
> > $  approximately lies in a low-dimensional subspace of $\mathbb{R}^{n \times n}$.
> > For example, in Fig. 2, we visualize the singular value decay of the matrix
> > $
> > \begin{bmatrix}
> > \operatorname{vec}(G(p_1)) & \cdots & \operatorname{vec}(G(p_m))
> > \end{bmatrix},
> > $where $G(p)$ denotes the matrix operation result for a real-world channel matrix at parameter $p$ and $\{p_1, \dots, p_m\}$ are sampled parameter values. The rapid decay of singular values confirms the presence of low-rank structure along the parameter dimension.
> >
> > To better validate the low-rank performance, we have added synthetic experiments with explicitly controlled parametric rank. The table below (i.e., Table 12 in the revised manuscript) reports the performance of NeuMatC compared to randomized SVD across different parametric ranks. Compared to randomized SVD, NeuMatC consistently achieves both higher accuracy and faster inference, as randomized SVD relies on the stronger assumption that each individual matrix is low-rank.
> >
> > We have provided the truncation rank settings used for randomized SVD in Appendix F, and included singular value decay plots for both real-world and synthetic datasets in Appendix G.
> >
> > **Table: Comparison of NeuMatC and RSVD on synthetic 512×512 parametric matrices
> > with controlled rank along the parameter dimension.**
> > The truncation rank of RSVD is set to 85.
> >
> > | Parametric Rank | NeuMatC CPU Time (ms) | NeuMatC RelErr | RSVD CPU Time (ms) | RSVD RelErr |
> > |------------------|------------------------|------------------|----------------------|--------------|
> > | 10               | 1.45 × 10⁰             | 3.90 × 10⁻³     | 23.572 × 10¹         | 1.23 × 10⁻² |
> > | 20               | 3.81 × 10⁰             | 2.70 × 10⁻³     | 24.074 × 10¹         | 1.15 × 10⁻² |
> > | 30               | 3.77 × 10⁰             | 4.61 × 10⁻³     | 23.902 × 10¹         | 1.32 × 10⁻² |

---

> > > ### Comment · Reviewer_WYpt · 2025-11-24
> > >
> > > Thank you for the detailed rebuttal and the additional experiments, which have clarified many of my initial concerns.
> > >
> > > I had initially assumed that your algorithm could be extended to large-scale eigenvalue problems, which represent a major computational bottleneck in industrial applications. However, based on your response, it seems this is not the intended scope.
> > >
> > > You have indeed proposed an end-to-end neural network for solving eigenvalue problems in the context of small- to medium-sized matrices. A key limitation of this data-driven approach is that once the network is trained, its inference error is fixed. It cannot be adjusted to meet the varying precision requirements of real-world applications, particularly the high spectral accuracy often demanded. Furthermore, the matrices considered in this paper are relatively small, for which traditional algorithms are already very fast, making the gains from acceleration marginal. In the domain of small- to medium-sized matrices, stability and accuracy are often more critical than speed. I personally see limited value in accelerating a method if its error cannot be guaranteed.
> > >
> > > Your work also appears to be limited to computing the full set of singular values and vectors, unlike traditional algorithms that can efficiently compute a partial spectrum (e.g., the largest k singular values) as needed.
> > >
> > > Regarding the experiment in Appendix H with matrices of size $10^5 \times 10^5$, the problem seems to be parameterized by only a very small number of parameters (apparently just one parameter, $p$). For such low-dimensional parameter spaces, we do not necessarily need a neural network. Simpler methods like symbolic regression or piecewise spline interpolation could be employed, offering guaranteed error bounds and potentially lower computational cost. The experiments I had hoped to see would address the highly time-consuming problems common in industrial settings, which typically involve: 1) very large matrix dimensions and 2) a very high-dimensional parameter space (e.g., >$10^5$ parameters). It appears the proposed algorithm is not designed to handle such scenarios.
> > >
> > > The comparison in Table 11 also seems problematic. Could you provide the data generation and training overhead for the main experiments, for instance, in a CPU-only environment? Additionally, could you cite literature demonstrating that industrial scenarios frequently involve repeatedly solving matrices that can be represented by a small set of parameters, as is the case in your experiments?
> > >
> > > If the workflow requires invoking traditional solvers to generate a dataset and then undergoing a separate training phase, all for a problem involving small- to medium-sized matrices or a low-dimensional parameter space, then from my perspective, the practical significance of this algorithm is quite low.
> > >
> > > Finally, concerning the handling of singular value crossings, a more concrete description of the alignment procedure would be beneficial. Otherwise, I am left to suspect that the algorithm's effectiveness might be largely due to the simplicity of the test cases, which may not feature such complexities.
> > >
> > > Based on these outstanding issues, particularly the very limited scope of applicability, I will maintain my current rating. I welcome further discussion on these points with the authors and other reviewers, and I am open to reconsidering my assessment if additional, relevant experiments are provided.

---

> > > > ### Author Response · Authors · 2025-12-02
> > > > **Response to Reviewer WYpt Round 2 (Part 1)**
> > > >
> > > > >**Q1.** Thank you for the detailed rebuttal and the additional experiments, which have clarified many of my initial concerns.
> > > >
> > > > **R1.** We thank the reviewer for the detailed follow-up and for acknowledging that our rebuttal clarified several concerns.
> > > >
> > > > >**Q2.** I had initially assumed that your algorithm could be extended to large-scale eigenvalue problems, which represent a major computational bottleneck in industrial applications. However, based on your response, it seems this is not the intended scope.
> > > >
> > > > >You have indeed proposed an end-to-end neural network for solving eigenvalue problems in the context of small- to medium-sized matrices. A key limitation of this data-driven approach is that once the network is trained, its inference error is fixed. It cannot be adjusted to meet the varying precision requirements of real-world applications, particularly the high spectral accuracy often demanded.
> > > >
> > > > **R2.** Thanks for the comments.
> > > > We would like to clarify that our work does not target the standard challenging large-scale computational tasks primarily arising in traditional scientific computing, but a different class of challenging tasks recently emerging in modern industrial systems (e.g., 5G/6G massive MIMO [1,2]). In these scenarios, **a large batch of moderate-sized matrices must be processed under strict latency constraints**. Consequently, the primary requirement is **high computational efficiency with acceptable accuracy**, rather than the high-precision accuracy pursued in traditional scientific computing problems.
> > > >
> > > > [1] Gong J, Kalia A, Yu M. Scalable distributed massive MIMO baseband processing. NSDI 2023.
> > > > [2] Ding J et al. Agora: Real-time massive MIMO baseband processing in software. CoNEXT 2020.
> > > >
> > > >
> > > > >**Q3.** Furthermore, the matrices considered in this paper are relatively small, for which traditional algorithms are already very fast, making the gains from acceleration marginal. In the domain of small- to medium-sized matrices, stability and accuracy are often more critical than speed. I personally see limited value in accelerating a method if its error cannot be guaranteed.
> > > >
> > > > **R3.** We agree that moderate-sized matrix operations are a highly mature domain, where achieving further acceleration over state-of-the-art numerical methods is particularly challenging. However, modern industrial applications (e.g., wireless communication and real-time signal processing) increasingly require much higher efficiency when processing *large batches* of such matrices. NeuMatC addresses this practical challenge from a continuous representation perspective which allows exploring the relations along the parameter dimension (i.e., inherent continuity and their relations in a low-rank structure). Numerical experiments on moderate-size parametric matrices demonstrate its advantages over state-of-the-art baselines while maintaining acceptable accuracy. Moreover, NeuMatC is also applicable to large-scale parametric matrix operation problems. As shown in the table below (i.e., Table 11 in the revised manuscript), NeuMatC achieves around $4\times$ speedup on large-scale parametric linear systems compared with a representative Krylov subspace recycling method.
> > > >
> > > > **Table: Comparison between Krylov subspace recycling and NeuMatC (parametric sparse linear systems, size $10^5 \times 10^5$, 200 systems)**
> > > >
> > > > | Method            | Data Preparation (ms) | Training (ms) | Inference (ms) | RelErr             | Speedup |
> > > > |-------------------|------------------------|---------------|----------------|--------------------|---------|
> > > > | Krylov Recycling  | ---                    | ---           | $5.19\times10^{5}$ | $1.12\times10^{-3}$ | $1.00\times$ |
> > > > | NeuMatC           | $1.2970\times10^{4}$   | $1.31\times10^{5}$ | $8.01$        | $1.11\times10^{-3}$ | $3.74\times$ |
> > > >
> > > > >**Q4.** Your work also appears to be limited to computing the full set of singular values and vectors, unlike traditional algorithms that can efficiently compute a partial spectrum (e.g., the largest k singular values) as needed.
> > > >
> > > > **R4.**  NeuMatC is a general framework that supports both full and partial SVD.
> > > > Given a parametric matrix $\mathbf{A}(p)\in\mathbb{R}^{m\times n}$, NeuMatC can directly learn a mapping from the parameter $p$ to the top-$k$ singular components.Specifically, we parameterize
> > > > $$
> > > > \tilde{\mathbf U}_k(p)=\mathcal{C}_U\times_3\Phi_U(p),\qquad
> > > > \tilde{\mathbf S}_k(p)=\mathcal{C}_S\times_3\Phi_S(p),\qquad
> > > > \tilde{\mathbf V}_k(p)=\mathcal{C}_V\times_3\Phi_V(p),
> > > > $$
> > > > where $\mathcal{C}_U\in\mathbb{R}^{m\times k\times d}$, $\mathcal{C}_V\in\mathbb{R}^{n\times k\times d}$, and $\mathcal{C}_S\in\mathbb{R}^{k\times 1\times d}$   are the learnable latent tensors. Thus, NeuMatC can directly output the partial SVD  $(\mathbf U_k(p), \mathbf S_k(p), \mathbf V_k(p))$ at any parameter $p$.

---

> > > > ### Author Response · Authors · 2025-12-02
> > > > **Response to Reviewer WYpt Round 2 (Part 2)**
> > > >
> > > > >**Q5.** Regarding the experiment in Appendix H with matrices of size $10^5 \times 10^5$, the problem seems to be parameterized by only a very small number of parameters (apparently just one parameter, $p$). For such low-dimensional parameter spaces, we do not necessarily need a neural network. Simpler methods like symbolic regression or piecewise spline interpolation could be employed, offering guaranteed error bounds and potentially lower computational cost. The experiments I had hoped to see would address the highly time-consuming problems common in industrial settings, which typically involve: 1) very large matrix dimensions and 2) a very high-dimensional parameter space (e.g., >10^5 parameters). It appears the proposed algorithm is not designed to handle such scenarios.
> > > >
> > > > **R5.**  The PDE experiment in Appendix H demonstrates the generality of NeuMatC on standard challenging large-scale problems arising in scientific computing. Moreover, NeuMatC is a general framework and naturally extends to multi-dimensional parameters by allowing $\Phi_\theta$ to take $p \in \mathbb{R}^d$ as input. We conduct a two-dimensional inversion experiment on synthetic $128\times128$ matrices generated over a $50\times50$ parameter grid. NeuMatC is trained on only 5 percent uniformly selected parameter points. As shown in the Table below, NeuMatC achieves low approximation error, demonstrating that NeuMatC remains effective in multi-dimensional settings with very sparse supervision. While symbolic regression or spline interpolation can fit simple parametric curves in low-dimensional spaces, they suffer from the curse of dimensionality as parameter dimensions grow.
> > > >
> > > > **Table: NeuMatC performance on a two-dimensional inversion task (128×128 matrices)**
> > > > Results are averaged over 20 independently generated synthetic datasets.
> > > >
> > > > | CPU Time (ms) | RelErr |
> > > > |---------------|-------------------------------------------|
> > > > | $3.7 \times 10^{-1}$ | $4.8 \times 10^{-4} \pm 2.4 \times 10^{-5}$ |
> > > >
> > > > ---
> > > >
> > > > >**Q6.** The comparison in Table 11 also seems problematic. Could you provide the data generation and training overhead for the main experiments, for instance, in a CPU-only environment? Additionally, could you cite literature demonstrating that industrial scenarios frequently involve repeatedly solving matrices that can be represented by a small set of parameters, as is the case in your experiments?
> > > >
> > > > **R6.**  Thanks for the comment. NeuMatC offers very fast inference, and even after including data preparation and training time, it can still provide clear overall runtime advantages when handling large batches of matrices.
> > > > The revised Table 11 (table below) shows that NeuMatC achieves approximately $4\times$ overall speedup on parametric sparse linear systems from PDEs, compared with a representative Krylov subspace recycling method on the CPU platform. These speedups are relevant in practical applications like wireless communication, where parametric matrices often depend on variables such as frequency, user location, and angles of arrival [1,2].
> > > >
> > > > **Table: Comparison between Krylov subspace recycling and NeuMatC (parametric PDE systems, size $10^5 \times 10^5$, 200 systems)**
> > > >
> > > > | Method            | Data Preparation (ms) | Training (ms) | Inference (ms) | RelErr              | Speedup |
> > > > |-------------------|------------------------|---------------|----------------|---------------------|---------|
> > > > | Krylov Recycling  | ---                    | ---           | $5.19\times10^{5}$ | $1.12\times10^{-3}$ | $1.00\times$ |
> > > > | NeuMatC           | $1.29\times 10^{4}$  | $1.31\times10^{5}$ | $8.01$        | $1.11\times10^{-3}$ | $3.74\times$ |

---

> ### Author Response · Authors · 2025-11-22
> **Response to Reviewer WYpt Part 3**
>
> > **W1.4 (Neglect of Relevant Literature):**
> > The paper fails to compare against a significant body of relevant work.
>
> ---
>
> > **W1.4.1:**
> > There is no direct comparison with other neural network-based eigenvalue/SVD solvers that share a similar PINN-like architecture (e.g., **neuralSVD**, **PMNN**, **STNet** [1–3]). A comparison of accuracy and speed against these methods is necessary to position the contribution.
>
> **R1.4.1:** NeuralSVD, PMNN, and STNet primarily target high-dimensional linear operator eigenvalue problem, which are fundamentally different from the problems considered in our work. NeuMatC is designed as a general framework for learning parametric matrix operations arising in real-world scenarios, such as matrix QR, inversion, and SVD. In the manuscript, we have compared NeuMatC against the state-of-the-art parametric matrix operation methods, including zeroing neural network-based methods (ZNNI for inversion [1] and ZNN-SVD for SVD [2]). As shown in Tables 1 and 2 of the manuscript, NeuMatC achieves significantly higher accuracy and much faster inference compared with ZNNI and ZNN-SVD, demonstrating the efficiency and generality of NeuMatC.
>
>  **[1]** Dimitrios Gerontitis et al.   *A novel extended Li zeroing neural network for matrix inversion.*
>  Neural Computing and Applications, 35(19):14129–14152, 2023.
>
>  **[2]** Jianrong Chen and Yunong Zhang.  *Online singular value decomposition of time-varying matrices via zeroing neural dynamics.*   Neurocomputing, 383:314–323, 2020.
>
> ---
>
> > **W1.4.2:**
> > More importantly, the paper ignores a rich field of traditional numerical methods designed specifically for series of related matrix problems. Techniques like **Krylov subspace recycling** [4], the **CHASE** algorithm [5], or using the invariant subspace from a previous solution as the initial guess for **LOBPCG** are designed to exploit the very same problem structure. A thorough comparison with these methods is essential to demonstrate the novelty and practical advantage of the proposed framework.
>
> **R1.4.2:** Thank you for the suggestion. Classical methods like Krylov subspace recycling, CHASE, and LOBPCG are tailored for accelerating specific operations (e.g., large-scale sparse linear systems and Hermitian eigenvalue problems). In contrast, NeuMatC targets efficiently performing general parametric matrix operations (including QR, inversion, and SVD) over a large number of moderate-sized dense matrices from real-world applications [3,4].
>
> To assess NeuMatC’s broader applicability, we include a comparison with a representative Krylov subspace recycling method on large-scale sparse linear systems from parametric PDE discretizations. As shown in the table below (i.e., Table 11 in the revised manuscript), NeuMatC achieves about **4× acceleration** in overall runtime while maintaining comparable accuracy.
>
> **Table: Comparison between Krylov subspace recycling and NeuMatC on parametric sparse linear systems
> arising from advection–diffusion–reaction PDEs.**  Each system has size $10^5 \times 10^5$, and RelErr is measured as  $\|\mathbf{A}(p)\mathbf{x}(p)-\mathbf{b}(p)\|_2 / \|\mathbf{b}(p)\|_2$.  Inference times correspond to solving 200 systems.
>
> | Method             | CPU Train (ms)   | CPU Inference (ms) | RelErr              | Speedup |
> |--------------------|------------------|----------------------|----------------------|---------|
> | Krylov Recycling   | —                | 5.19 × 10⁵           | 1.12 × 10⁻³          | 1.00×   |
> | NeuMatC            | 1.31 × 10⁵       | 8.01 × 10⁰           | 1.11 × 10⁻³          | 3.96×   |
>
>  **Note.**   Krylov Recycling results are obtained using the GCRO-DR algorithm, with implementation adapted from the codes developed in [3].   All methods are implemented under the same Python environment.  For GCRO-DR, we employ an ILU(0) preconditioner and use the previous solution as the initial guess for the next parameter instance.
>
>  **[3]** A. de Sturler.   *Recycling Krylov subspaces for sequences of linear systems.*   SIAM Journal on Scientific Computing, 28(5):1651–1674, 2006.

---

> ### Author Response · Authors · 2025-11-22
> **Response to Reviewer WYpt Part 4**
>
> > **W2: Limited Scope and Scalability Analysis:**
> > The experiments are conducted in a narrow and relatively easy regime, which limits the generalizability of the findings.
>
> ---
>
> > **W2.1 (Small Matrix Sizes):**
> > The experiments are confined to relatively small matrices (dimension <10,000). These problems can often be solved in seconds or less by traditional methods. The true challenge in industrial and scientific applications lies in large-scale matrices (e.g., dimensions from 10⁴ to 10⁶). The paper fails to demonstrate the method’s performance in this critical regime. Comparisons should be made against state-of-the-art iterative solvers like **Krylov-Schur**, **LOBPCG**, and **JD** (e.g., using implementations from **SLEPc** [6]) for such large-scale problems.
>
> > **W2.2 (Limited Matrix Types):**
> > The evaluation should consider both **sparse matrices** (common in PDE discretizations) and **dense matrices** (e.g., from quantum chemistry). The paper’s focus on low-rank matrices from communication is a significant limitation. The performance on non-low-rank matrices, such as those arising from PDE stability analysis (e.g., modal analysis of Maxwell’s equations), is not explored. If the method is only applicable to low-rank problems, this limitation must be clearly stated in the abstract and introduction.
>
> **R2:** NeuMatC does not focus on large-scale sparse matrix computations typically solved by traditional methods such as LOBPCG, JD, and Krylov-Schur. Instead, we focus on efficiently performing parametric matrix operations on a large number of **moderate-sized and dense** matrices, which frequently arises in applications such as wireless communication and signal processing [4,5]. Although NeuMatC is primarily designed for moderate-sized dense matrices, we further evaluate NeuMatC on large-scale sparse linear systems from parametric PDEs. In comparison with a representative Krylov subspace recycling method, NeuMatC achieves about **4× total runtime speedup** while maintaining comparable accuracy, as shown in Table below  (i.e., Table 11 in the revised manuscript).
>
> As for low-rankness, NeuMatC is designed as a general framework and does not require the low-rankness assumptions of individual matrix $A(p)$ and its output $G(p)$. Instead, we assume a milder condition commonly observed in real-world applications, i.e., **low-rankness along the parameter dimension**. That is, the collection $\{G(p): p \in \mathcal{P}\}$ approximately lies in a low-dimensional subspace of $\mathbb{R}^{n \times n}$. For example, in Fig. 2, we compute singular values on matrices $
> \begin{bmatrix}
> \operatorname{vec}(G(p_1)) & \cdots & \operatorname{vec}(G(p_m))
> \end{bmatrix},$  where $G(p)$ denotes the matrix operation result for a real-world channel matrix at parameter $p$ and $\{p_1,\dots,p_m\}$ are sampled parameter values. The rapid decay of singular values indicates low-rankness along the parameter dimension. We have revised the manuscript to clarify this point.
>
> The individual matrix structures (e.g., sparsity or low-rankness) can be explored for further acceleration in the future.
>
> **Table: Comparison between Krylov subspace recycling and NeuMatC on parametric sparse linear systems
> arising from advection–diffusion–reaction PDEs.**  Each system has size $10^5 \times 10^5$, and RelErr is measured as  $\|\mathbf{A}(p)\mathbf{x}(p)-\mathbf{b}(p)\|_2 / \|\mathbf{b}(p)\|_2$.  Inference times correspond to solving 200 systems.
>
> | Method             | CPU Train (ms)   | CPU Inference (ms) | RelErr              | Speedup |
> |--------------------|------------------|----------------------|----------------------|---------|
> | Krylov Recycling   | —                | 5.19 × 10⁵           | 1.12 × 10⁻³          | 1.00×   |
> | NeuMatC            | 1.31 × 10⁵       | 8.01 × 10⁰           | 1.11 × 10⁻³          | 3.96×   |
>
>
>  **[4]** Chanho Jeon, Zhenhao Li, and Christoph Studer.   *Approximate gram-matrix interpolation for wideband massive MU-MIMO systems.*  >IEEE Transactions on Vehicular Technology, 69(5):4677–4688, 2020.
>
>  **[5]** Jiacheng Gong, Anirudh Kalia, and Minjie Yu.  *Scalable distributed massive MIMO baseband processing.*   In *Proceedings of the 20th USENIX Symposium on Networked Systems Design and Implementation (NSDI)*, pages 405–417, 2023.

---

> ### Author Response · Authors · 2025-11-22
> **Response to Reviewer WYpt Part 5**
>
> > **W3: Inadequate Justification of Low-Rank Claims:**
> > While the paper is motivated by the low-rank property of parametric matrices, the experimental validation is superficial.
> > - The authors do not provide details on the rank structure of their test matrices (e.g., a plot of singular value decay or loss versus rank). This makes it difficult to assess how “low-rank” the problems truly are and how well the method exploits this property.
> > - To properly validate the low-rank performance, the authors should construct synthetic datasets with explicitly controlled ranks and compare against specialized low-rank SVD algorithms. Simply testing on real-world datasets is insufficient.
> > - When comparing to randomized SVD, crucial details like the choice of the compression dimension (rank) are omitted. The performance of randomized SVD is highly dependent on this parameter, which should be chosen based on the target rank or error tolerance [7].
>
> **R3:** Thank you for the comment. Traditional low-rank methods such as randomized SVD typically rely on a strong assumption that each individual matrix is low-rank. In contrast, NeuMatC does not assume low-rankness of the individual matrices. We adopt a *milder* assumption commonly observed in real-world applications, i.e., **low-rankness along the parameter dimension**, where the collection  $\{G(p): p \in \mathcal{P}\}$  approximately lies in a low-dimensional subspace of $\mathbb{R}^{n \times n}$.  For example, in Fig. 2, we visualize the singular value decay of the matrix  $
> \begin{bmatrix}
> \operatorname{vec}(G(p_1)) & \cdots & \operatorname{vec}(G(p_m))
> \end{bmatrix},
> $ where $G(p)$ denotes the matrix operation result for a real-world channel matrix at parameter $p$, and $\{p_1,\dots,p_m\}$ are sampled parameter values. The rapid decay of singular values confirms the presence of low-rank structure along the parameter dimension.
>
> To better validate the low-rank performance, we have added synthetic experiments with explicitly controlled parametric rank. The table below (i.e., Table 12 in the revised manuscript) reports the performance of NeuMatC compared to randomized SVD across different parametric ranks. Compared to randomized SVD, NeuMatC consistently achieves both higher accuracy and faster inference, as randomized SVD relies on the stronger assumption that each individual matrix is low-rank.
>
> We have provided the truncation rank settings used for randomized SVD in Appendix F, and included singular value decay plots for both real-world and synthetic datasets in Appendix G.
>
> **Table: Comparison of NeuMatC and RSVD on synthetic 512×512 parametric matrices
> with controlled rank along the parameter dimension.**
> The truncation rank of RSVD is set to 85.
>
> | Parametric Rank | NeuMatC CPU Time (ms) | NeuMatC RelErr | RSVD CPU Time (ms) | RSVD RelErr |
> |------------------|------------------------|------------------|----------------------|--------------|
> | 10               | 1.45 × 10⁰             | 3.90 × 10⁻³     | 23.572 × 10¹         | 1.23 × 10⁻² |
> | 20               | 3.81 × 10⁰             | 2.70 × 10⁻³     | 24.074 × 10¹         | 1.15 × 10⁻² |
> | 30               | 3.77 × 10⁰             | 4.61 × 10⁻³     | 23.902 × 10¹         | 1.32 × 10⁻² |

---

> ### Author Response · Authors · 2025-11-22
> **Response to Reviewer WYpt Part 6**
>
> > **Q1 (Handling of Singular Value/Vector Crossings):**
> > A critical concern is the handling of singular value/vector crossings. As a parameter $p$ varies continuously, the ordering of singular values can change (e.g., the first and second singular values swap). A neural network trained to output the “k-th” singular vector would likely fail at such crossing points, as it would need to abruptly switch its output from one vector function to another, which is a discontinuous behavior. How does the proposed framework handle this well-known phenomenon in parametric eigenvalue problems? Were the datasets specifically chosen to avoid such crossings? A discussion of this issue can be found in literature such as [1] P376.
>
> **R1:** NeuMatC is designed as a general framework for learning parametric matrix operators and does not introduce specific designs to explicitly handle singular value/vector crossings. For operations such as SVD, where such crossings may cause discontinuities along the parameter dimension, we apply an alignment procedure on the training data following [1] to enforce continuity across parameter samples. We have made this detail more prominent in the revised manuscript.
>
>  **[1]** Mahdi Tohidian, Hamidreza Amindavar, and Ali M. Reza.   *A DFT-based approximate eigenvalue and singular value decomposition of polynomial matrices.*   EURASIP Journal on Advances in Signal Processing, 2013(93):1–11, 2013.
>
> ---
>
> > **Q2 (Scalability and Fair Baselines):**
> > Could the authors provide performance benchmarks on truly large-scale matrices (e.g., dimensions from $10^4$ to $10^6$), for both sparse and dense cases? In this context, could you compare against established large-scale iterative solvers like **Krylov-Schur**, **LOBPCG**, and **JD** (e.g., from **SLEPc**), including fair comparisons on GPU (e.g., vs. **cuSOLVER**) and with appropriate **preconditioning**?
>
> **R2:** The mentioned methods (e.g., LOBPCG, JD, Krylov-Schur) are classical solvers for large-scale *sparse* problems which exploit sparsity structures. In contrast, NeuMatC targets a different task, i.e., efficiently performing parametric matrix operations over a large number of *moderate-sized, dense* matrices, as commonly required in applications like wireless communication and signal processing [4,5]. To address this task, we propose NeuMatC, which is a general framework whose learned model can run efficiently on both CPU and GPU. While NeuMatC is primarily designed for moderate-sized dense problems, we further evaluate its efficiency on large-scale sparse linear systems of size $10^5 \times 10^5$ from parametric PDEs. We compare NeuMatC against a representative Krylov subspace recycling method (GCRO-DR), using ILU(0) preconditioning and initializing each system with the solution from the previous parameter. As shown in Table below (i.e., Table 11 in the revised manuscript), NeuMatC achieves about 4× total runtime speedup for 200 systems while maintaining comparable accuracy.
>
> **Table: Comparison between Krylov subspace recycling and NeuMatC on parametric sparse linear systems
> arising from advection–diffusion–reaction PDEs.**  Each system has size $10^5 \times 10^5$, and RelErr is measured as  $\|\mathbf{A}(p)\mathbf{x}(p)-\mathbf{b}(p)\|_2 / \|\mathbf{b}(p)\|_2$.  Inference times correspond to solving 200 systems.
>
> | Method             | CPU Train (ms)   | CPU Inference (ms) | RelErr              | Speedup |
> |--------------------|------------------|----------------------|----------------------|---------|
> | Krylov Recycling   | —                | 5.19 × 10⁵           | 1.12 × 10⁻³          | 1.00×   |
> | NeuMatC            | 1.31 × 10⁵       | 8.01 × 10⁰           | 1.11 × 10⁻³          | 3.96×   |
>
>
> While our main experiments focus on CPU, we have additionally included comparisons with GPU-accelerated solvers (e.g., cuSOLVER) for completeness. As shown in the table below (i.e., Table 13 in the revised manuscript), NeuMatC achieves substantial speedups over cuSOLVER on both inversion and SVD tasks, further demonstrating its efficiency.
>
> **Table: Runtime comparison of NeuMatC and cuSOLVER for batched matrix inversion and SVD on GPU.**
>
> | Matrix Size | Batch Size | cuSOLVER (ms) – Inversion | NeuMatC (ms) – Inversion | Speedup | cuSOLVER (ms) – SVD | NeuMatC (ms) – SVD | Speedup |
> |-------------|------------|----------------------------|---------------------------|---------|----------------------|---------------------|---------|
> | 32×32 | 128 | 2.7 × 10⁻¹ | 2.3 × 10⁻¹ | 1.17× | 9.3 × 10⁻¹ | 5.3 × 10⁻¹ | 1.77× |
> |  | 256 | 2.9 × 10⁻¹ | 2.0 × 10⁻¹ | 1.49× | 1.29 × 10⁰ | 5.5 × 10⁻¹ | 2.35× |
> |  | 512 | 4.2 × 10⁻¹ | 2.6 × 10⁻¹ | 1.62× | 2.33 × 10⁰ | 5.9 × 10⁻¹ | 3.98× |
> | 512×512 | 128 | 2.47 × 10¹ | 7.8 × 10⁻¹ | 31.71× | 3.31 × 10³ | 1.44 × 10⁰ | 2295.04× |
> |  | 256 | 5.48 × 10¹ | 1.22 × 10⁰ | 44.87× | 6.64 × 10³ | 2.80 × 10⁰ | 2374.41× |
> |  | 512 | 1.15 × 10² | 2.28 × 10⁰ | 50.36× | 1.35 × 10⁴ | 4.69 × 10⁰ | 2882.23× |

---

> ### Author Response · Authors · 2025-11-22
> **Response to Reviewer WYpt Part 7**
>
> > **Q3: Practical Applicability and Overhead**
> > The proposed method requires a pre-generated training set of related matrices.
>
> > **Q3.1:**
> > In a real-world scenario where matrix problems may arise independently and without a unified parametric representation, what is the applicable scope of this method?
>
> **R3.1:** Thank you for the question. NeuMatC does not rely on the existence of an explicit parametric representation. When the parametric representation is unavailable, NeuMatC can still learn the mapping from parameters to matrix operation results directly from a few supervised samples. This makes NeuMatC applicable to a wide range of real-world applications such as wireless communication and signal processing.
>
> ---
>
> > **Q3.2:**
> > When comparing against traditional solvers that work on a single instance, should the overhead of data generation and network training be included in the timing? What are the specific application scenarios where generating such a large, correlated dataset is feasible and the amortization of training cost makes this approach superior to traditional methods that can exploit problem-to-problem correlation without pre-training (e.g., **Krylov subspace recycling** [1], the **CHASE** algorithm [2], or using the invariant subspace from a previous solution as the initial guess for **LOBPCG**)?
>
> **R3.2:** Thanks for the comment. NeuMatC targets the computational challenges of parametric matrix operations, which commonly arise in real-world applications such as wireless communication systems and often require solving thousands of matrices efficiently [1,2]. For such tasks, NeuMatC learns the mapping using only a small number of supervised samplings, enabling effective amortization of the training cost across large workloads. As shown in Table below  (i.e., Table 11 in the revised manuscript), we report the overall runtime of NeuMatC (i.e., training and inference time) for large-scale sparse linear systems from parametric PDEs. We observe that NeuMatC achieves about **4× overall speedup** over the representative Krylov subspace recycling method when processing 200 systems of size $10^5 \times 10^5$. These results demonstrate NeuMatC’s efficiency even when training cost is taken into account.
>
> **Table: Comparison between Krylov subspace recycling and NeuMatC on parametric sparse linear systems
> arising from advection–diffusion–reaction PDEs.**  Each system has size $10^5 \times 10^5$, and RelErr is measured as  $\|\mathbf{A}(p)\mathbf{x}(p)-\mathbf{b}(p)\|_2 / \|\mathbf{b}(p)\|_2$.  Inference times correspond to solving 200 systems.
>
> | Method             | CPU Train (ms)   | CPU Inference (ms) | RelErr              | Speedup |
> |--------------------|------------------|----------------------|----------------------|---------|
> | Krylov Recycling   | —                | 5.19 × 10⁵           | 1.12 × 10⁻³          | 1.00×   |
> | NeuMatC            | 1.31 × 10⁵       | 8.01 × 10⁰           | 1.11 × 10⁻³          | 3.96×   |

---

> ### Author Response · Authors · 2025-12-02
> **Response to Reviewer WYpt Round 2 (Part 3)**
>
> >**Q7.** If the workflow requires invoking traditional solvers to generate a dataset and then undergoing a separate training phase, all for a problem involving small- to medium-sized matrices or a low-dimensional parameter space, then from my perspective, the practical significance of this algorithm is quite low.
>
> **R7.**  Thanks for the comment. NeuMatC offers very fast inference, and even after including data preparation and training time, it can still provide clear overall runtime advantages when handling large batches of moderate-sized matrices. For example, in the table below (Table 5 in the revised manuscript), we observe that NeuMatC achieves up to $28.75\times$ overall speedup over the state-of-the-art cuSOLVER for batched SVD of moderate-sized matrices. The acceleration becomes even more pronounced as the batch size increases.
>
> **Table: Overall runtime of NeuMatC for SVDs of 512×512 matrices (GPU platform)**
> Results averaged over 20 synthetic datasets. NeuMatC: 500 epochs, relative error ≈ $1.0\times10^{-3}$.
>
> | Batch size | cuSOLVER (ms) | Data Prep (ms) | Train (ms) | Inference (ms) | Speedup |
> |------------|----------------|----------------|------------|----------------|---------|
> | 512        | $1.33\times10^{4}$ | $1.04\times10^{3}$ | $6.22\times10^{3}$ | $1.30$ | $2.03\times$ |
> | 1024       | $2.62\times10^{4}$ | $1.04\times10^{3}$ | $6.22\times10^{3}$ | $1.97$ | $3.98\times$ |
> | 2048       | $1.79\times10^{5}$ | $1.04\times10^{3}$ | $6.22\times10^{3}$ | $3.41$ | $26.90\times$ |
>
> ---
>
> >**Q8.** Finally, concerning the handling of singular value crossings, a more concrete description of the alignment procedure would be beneficial. Otherwise, I am left to suspect that the algorithm's effectiveness might be largely due to the simplicity of the test cases, which may not feature such complexities.
>
> **R8.**  Thanks for the comments. Our work is motivated by practical parametric matrix problems arising in real-world applications such as wireless communication and signal processing. In our experiments, we evaluate NeuMatC on the widely used DeepMIMO wireless communication dataset, ensuring that the method is assessed under realistic parameter variations.   To properly handle singular-value crossings, we apply an alignment procedure following Tohidian et al. (2013) [3], using maximum-correlation matching with permutation and phase correction. This procedure is described in Appendix J of the revised manuscript.
>
> [3] Mahdi Tohidian, Hamidreza Amindavar, and Ali M. Reza.  A DFT-based approximate eigenvalue and singular value decomposition of polynomial matrices.  *EURASIP Journal on Advances in Signal Processing*, 2013(93):1–11, 2013.

---

### Official Review · Reviewer_8kXx · 2025-10-30

**Soundness:** 4
**Presentation:** 3
**Contribution:** 3
**Rating:** 8
**Confidence:** 4

**Summary:**

The authors train a MLP with a tensor head to solve matrix problems that are related in a 1 dimensional parameter space
by minimizing a data fidelity and structure consistency loss.

**Strengths:**

* The authors tackle a high impact problem as matrix operations are ubiquitous in science applications. Also the
  approach seems quite novel (when compared to the related work).
* The authors show strong results against classical numerical algorithms which are hard to beat. The results contain
  comprehensive metrics on test relative error, GFLOPs and runtime.
* The authors provide several theoretical claims that support their choice of mapping function.

**Weaknesses:**

* Having a method that only applies to 1 dimensional parameters seems limiting. Would the objective in Eq (3) be able to
  generalize to larger dimensions?
* No code is shared through a anonymized link. How are the authors planning to promote adoption and reproducibility?
Indeed there are some details in Appendix F, though the details are about ranges of hyperparameters.
* The speed-up improvements do not account for the NN training. Imagine that you have 100 parameters points where you
  want to run a matrix operations. How much more competitive is your method if you were to account for the MLP training?
  If your loss already requires $N_s = 100$ then there wouldn't be any improvement.

**Questions:**

* Lines 231, 245. Could you expand on the definition of "dense samples"?
* How do you choose between $N_s$ and $N_c$? It seems that $N_c$ should always be much larger than $N_s$ as we avoid
  having to create data with a numerical solver?
* What would happen if I pass all the unsupervised $N_c$ points to the Data Fidelity Loss? Would the trained model
  always be better? Or I guess you want the backprop to consider the structure so that the models is aware of that?
  Do the matrices have to be exactly the same size or can they vary? Imagine that I have the classical least squares solution $(X^T X + \alpha I)^{-1} X^Ty$ where $\alpha$ is a single
  dimensional parameter. Would your method work for d=3 coefficients and for d=5?
* Besides the Lipschitz constant, is there any intuition on which activations would work better for these type of matrix
  problems?

---

> ### Author Response · Authors · 2025-11-22
> **Response to Reviewer 8kXx Part 1**
>
> We sincerely thank all reviewers for their positive comments (e.g., “quite novel”, “interesting and novel approach”, and “promising idea”) and their constructive suggestions. Our study addresses the emerging challenge of efficiently performing a large number of parametric matrix operations on moderate-sized matrices, which frequently arises in applications such as wireless communication and signal processing.
>
> To address this challenge, we propose NeuMatC, a general framework for accelerating parametric matrix operations. Specifically, NeuMatC learns an efficient mapping from parameters to matrix operation results by exploring the relations along the parameter dimension (i.e., inherent continuity and their relations in a low-rank structure). Numerical experiments demonstrate advantages over state-of-the-art baselines, achieving over $3\times$ speedup for parametric inversion and $10\times$ for parametric SVD in wireless communication tasks while maintaining good accuracy.
>
> For easy reference, a list of responses to the reviewers' comments has been compiled, and the corresponding changes have been highlighted in **blue** in the main text for easy reference of our revision. Should you need further information, please let us know. We look forward to hearing from you soon.
>
> > **W1: Having a method that only applies to 1 dimensional parameters seems limiting. Would the objective in Eq (3) be able to generalize to larger dimensions?**
>
> **R1:** Thanks for the valuable  comment.
> Our proposed NeuMatC is a general framework and can naturally generalize to multi-dimensional cases by extending
> $\Phi_\theta$ to take $p \in \mathbb{R}^d$ as input, leading to
> \[
> \mathbf{G}(p) = \mathcal{C} \times_3 \Phi_\theta(p), \qquad p \in \mathbb{R}^d.
> \]
> where $\Phi_\theta: \mathbb{R}^d \to \mathbb{R}^r$ is an MLP.
> We conduct a two-dimensional inversion experiment on synthetic $128\times128$ matrices generated over a $50\times 50$ parameter grid. NeuMatC is trained using $5\%$ uniformly selected points and tested on 100 points. As shown Table below (i.e., Table 6 in the revised manuscript), we observe that NeuMatC achieves low approximation error and fast inference, demonstrating the effectiveness of NeuMatC in multi-dimensional case.
>
> **Table: NeuMatC performance on a two-dimensional inversion task (128×128 matrices).
> Results are averaged over 20 independently generated synthetic datasets.**
>
> | CPU Time (ms)        | RelErr                                |
> |----------------------|----------------------------------------|
> | 3.7 × 10⁻¹           | 4.8 × 10⁻⁴ ± 2.4 × 10⁻⁵                |

---

> ### Author Response · Authors · 2025-11-22
> **Response to Reviewer 8kXx Part 2**
>
> > **W2: No code is shared through an anonymized link. How are the authors planning to promote adoption and reproducibility? Indeed there are some details in Appendix F, though the details are about ranges of hyperparameters.**
>
> **R2:** Thanks for the comment. For reproducibility, we will release the code after the possible acceptance of the manuscript.
>
>
> > **W3: The speed-up improvements do not account for the NN training. Imagine that you have 100 parameters points where you want to run a matrix operations. How much more competitive is your method if you were to account for the MLP training? If your loss already requires $N_s = 100$ then there wouldn't be any improvement.**
>
> **R3:** Thanks for the comment. NeuMatC targets the computational challenges of parametric matrix operations, which commonly arise in real-world applications such as wireless communication systems and often require solving thousands of matrices efficiently~[1,2].
> For such tasks, NeuMatC learns the mapping using only a small number of sampled parameter points ($N_s \ll N_q$), enabling effective amortization of the training cost across large workloads.
> In the revised manuscript, we report the overall runtime of NeuMatC (i.e., training and inference time) for SVD tasks in Table below (i.e., Table 5 in the revised manuscript). We observe that NeuMatC achieves a 28.75$\times$ overall speedup over the high-performance GPU library cuSOLVER when processing 2048 matrices of size $512 \times 512$. These results demonstrate NeuMatC’s efficiency even when training cost is taken into account.
>
> **Table: Overall runtime of NeuMatC (including training and inference) for SVDs of 512×512 matrices on the GPU platform.  Results are averaged over 20 synthetic datasets. NeuMatC is trained for 500 epochs, reaching a relative error of approximately 1.0×10⁻³.**
>
> | Batch size | cuSOLVER (ms)       | NeuMatC Train (ms) | NeuMatC Inference (ms) | Speedup      |
> |------------|----------------------|---------------------|--------------------------|--------------|
> | 512        | 1.33×10⁴             | 6.22×10³            | 1.30                     | 2.15×        |
> | 1024       | 2.62×10⁴             | 6.22×10³            | 1.97                     | 4.22×        |
> | 2048       | 1.79×10⁵             | 6.22×10³            | 3.41                     | 28.75×       |
>
>
>  **[1]** Chanho Jeon, Zhenhao Li, and Christoph Studer.   Approximate gram-matrix interpolation for wideband massive MU-MIMO systems.   *IEEE Transactions on Vehicular Technology*, 69(5):4677–4688, 2020.
>
>  **[2]** Jiacheng Gong, Anirudh Kalia, and Minjie Yu.   Scalable distributed massive MIMO baseband processing.   In *Proceedings of the 20th USENIX Symposium on Networked Systems Design and Implementation (NSDI)*, pages 405–417, 2023.

---

> ### Author Response · Authors · 2025-11-22
> **Response to Reviewer 8kXx Part 3**
>
> > **Q1: Lines 231, 245. Could you expand on the definition of “dense samples”?**
>
> **R1:** In the revised manuscript, we clarify that “dense samples” refers to a high sampling rate of supervised samples, each of which requires expensive computation of ground-truth matrix operation results. To tackle this issue, NeuMatC introduces structure consistency loss to guide training, allowing NeuMatC to rely on only a small number of supervised samples.
>
>
> > **Q2: How do you choose between $N_s$ and $N_c$? It seems that $N_c$ should always be much larger than $N_s$ as we avoid having to create data with a numerical solver?**
>
> **R2:** Thank you for the question. In NeuMatC, we keep $N_s$ small to limit the cost of computing ground-truth matrix operations. To complement this, we introduce a structure consistency loss on a set of $N_c$ unsupervised collocation points. However, using too many collocation points can substantially increase training time while only marginally improving performance. We therefore adopt an adaptive sampling strategy that selects a compact but informative subset of collocation points. In our experiments, we set $N_s \in \{20, 40, 60, 80\}$ and the initial $N_c \in \{20, 80, 140, 200\}$, as detailed in Appendix F.

---

> ### Author Response · Authors · 2025-11-22
> **Response to Reviewer 8kXx Part 4**
>
> > **Q3.1: What would happen if I pass all the unsupervised $N_c$ points to the Data Fidelity Loss? Would the trained model always be better? Or I guess you want the backprop to consider the structure so that the models are aware of that?**
>
> **R3.1:** Sorry for the confusion. Passing all of the unsupervised $N_c$ points into the data-fidelity loss still allows us to apply the lightweight structure-consistency loss on the same set, enabling the model to remain structure-aware. Since this provides more supervision, the performance would indeed improve, but it would also increase training cost.
>
>
> > **Q3.2: Do the matrices have to be exactly the same size or can they vary?**
>
> **R3.2:** Our work is motivated by practical needs in real-world applications such as wireless communication and signal processing~[1,2], where the size of the involved parametric matrices remains fixed across the parameter domain. Variable-sized matrices can be explored in the future if there are real-world application demands.
>
>
> > **Q3.3: Imagine that I have the classical least squares solution $(X^{T}X + \alpha I)^{-1}X^{T}y$ where $\alpha$ is a single dimensional parameter. Would your method work for $d=3$ coefficients and for $d=5$?**
>
> **R3.3:** Thanks for the valuable comment. Our proposed NeuMatC is a general framework and can naturally generalize to multi-dimensional cases by extending
> $\Phi_\theta$ to take $p \in \mathbb{R}^d$ as input, leading to
> $$
> \mathbf{G}(p) = \mathcal{C} \times_3 \Phi_\theta(p), \qquad p \in \mathbb{R}^d.
> $$
> where $\Phi_\theta: \mathbb{R}^d \to \mathbb{R}^r$ is an MLP.  In the revised manuscript, we discuss the generalization to higher-dimensional parameter domains and present experiments demonstrating the effectiveness of NeuMatC in multi-dimensional cases. Please refer to the Discussion section for details.

---

> ### Author Response · Authors · 2025-11-22
> **Response to Reviewer 8kXx Part 5**
>
> > **Q4: Besides the Lipschitz constant, is there any intuition on which activations would work better for these type of matrix problems?**
>
> **R4:** Thank you for the question. We adopt the sine activation function inspired by its success in implicit neural representations, which can help us effectively capture complex structures [3]. In our numerical experiments, we conduct an ablation study comparing sine with widely used activations such as Sigmoid, GELU, ReLU, and Tanh. As shown in the table below (i.e., Table 9 in the revised manuscript), sine achieves the lowest relative error, demonstrating the advantage of sine in our NeuMatC framework.
>
> **Table: Impact of activation function on NeuMatC.**
>
> | Activations | Sigmoid | GELU | ReLU | Tanh | Sine |
> |-------------|---------|------|------|------|------|
> | RelErr (↓)  | 3.0 × 10⁻² | 1.5 × 10⁻¹ | 4.6 × 10⁻² | 2.6 × 10⁻² | **8.3 × 10⁻³** |
>
>  **[3]** Vincent Sitzmann, Julien N. P. Martel, Alexander W. Bergman, David B. Lindell, and Gordon Wetzstein.   Implicit neural representations with periodic activation functions.  In *Advances in Neural Information Processing Systems (NeurIPS)*, volume 33, pages 7462–7473, 2020.

---

### Meta-Review · Area_Chair_UANR · 2026-01-06

**Summary:**

Reviewers saw NeuMatC as a novel and potentially useful way to amortize parametric matrix operations over many instances, but concerns that drove the decision centered on: (i) whether the experiments and baselines are fair and representative of strong practice (GPU batched libraries, optimized implementations, and classical “sequence” methods), (ii) whether the method’s benefits matter in realistic settings given that it targets mainly small-to-medium matrices and produces a fixed-accuracy surrogate after training, and (iii) robustness issues for SVD (crossings/non-uniqueness) and the interpretation of the “low-rank” assumption, especially for inversion. The rebuttal strengthened the paper with additional experiments and clarifications, but it did not clearly change the stance of the strongest reject reviewer, and fully resolving the remaining objections would likely require broadening the contribution beyond the current problem framing (e.g., adding a mechanism to trade computation for higher accuracy at test time, or providing stronger evidence that the targeted workloads are common and cannot be handled comparably well by simpler surrogates).

**Reviewer Concerns:**

Addressed and generally convincing without major changes to the method:

-  Added GPU batched comparisons vs cuSOLVER and a C++ comparison vs Eigen; clarified the evaluation protocol over batches. This substantially improves the credibility of speed claims.

- Added end-to-end timing (training + inference) showing amortization benefits for large batch sizes. This is convincing for workloads with many matrices and repeated use, but depends on the realism of the assumed workload sizes.

- Clarified that “low-rank” refers to structure across the parameter dimension rather than per-matrix low-rank, and added singular-value decay evidence plus controlled-rank synthetic tests and RSVD rank settings. This addresses confusion about inversion and makes the assumptions clearer, though it still leaves questions about how broadly the assumption holds across applications.

-  Added an alignment procedure (permutation/phase correction via maximum-correlation matching) with appendix details. This is a plausible practical workaround, but it does not provide strong guarantees and may not satisfy reviewers expecting robustness across challenging crossing regimes.

- Added results on additional operations (QR/Cholesky/matrix exponential), supporting the “framework” claim.

Still outstanding or not persuasive enough to flip key rejects (would likely require substantive extensions or stronger evidence):

-  A reject reviewer argued the method cannot meet varying precision requirements because, once trained, inference error is essentially fixed; traditional solvers can increase accuracy by spending more computation. The rebuttal argues that “acceptable accuracy” is sufficient for latency-critical wireless/real-time settings, which is coherent but does not address applications where controllable accuracy is essential. Addressing this would likely require a hybrid design (e.g., neural initialization plus iterative refinement or explicit error control).

- The same reviewer questions whether accelerating small-to-medium matrices is meaningful because classical methods are already fast and stability/accuracy often matter more than speed. This is not a minor experimental gap; convincing such a reviewer would require stronger evidence of downstream impact and workload prevalence in the intended domain.

-  The reviewer suggested spline/polynomial/rational fitting or symbolic regression could replace a neural model when parameters are low-dimensional. The rebuttal claims fixed bases are insufficient and adds a polynomial vs NN comparison, but the discussion record does not show this concern being convincingly settled.

**Reviewer Scores:**

Reviewer 8kXx: likely unchanged at 8.

Reviewer WYpt: unchanged at 2 (explicitly maintained rating; remaining objections concern fixed-accuracy limitation and the perceived limited practical value, which would require substantive extensions to address).

Reviewer SGgi: likely increases from 4 to 6 (most concerns were directly clarified and supported with added experiments).

Reviewer 4t5r: unchanged at 4 (several points were resolved and additional comparisons were added), but there is no clear evidence they would move to accept; it depends on how convincing they find the added DLA and surrogate-baseline comparisons.

---

### Decision · Program_Chairs · 2026-01-26

Reject